# Neurons dispose of hyperactive kinesin into glial cells for clearance

Chao Xie[1,2,3,4,5,9], Guanghan Chen [1,2,3,4,5,9], Ming Li [1,2,3,4,5], Peng Huang[1,2,3,4,5], Zhe Chen [1,2,3,4,5], Kexin Lei[1,2,3,4,5], Dong Li[6,7], Yuhe Wang[1,2,3,4,5], Augustine Cleetus[8], Mohamed AA Mohamed[8], Punam Sonar[8], Wei Feng [6,7], Zeynep Ökten [8] & Guangshuo Ou [1,2,3,4,5 ✉]

## Abstract

**Microtubule-based kinesin motor proteins are crucial for intracellular transport, but their hyperactivation can be detrimental for cellular functions. This study investigated the impact of a constitutively active ciliary kinesin mutant, OSM-3CA, on sensory cilia in *C. elegans*. Surprisingly, we found that OSM-3CA was absent from cilia but underwent disposal through membrane abscission at the tips of aberrant neurites. Neighboring glial cells engulf and eliminate the released OSM-3CA, a process that depends on the engulfment receptor CED-1. Through genetic suppressor screens, we identified intragenic mutations in the OSM-3CA motor domain and mutations inhibiting the ciliary kinase DYF-5, both of which restored normal cilia in OSM-3CA-expressing animals. We showed that conformational changes in OSM-3CA prevent its entry into cilia, and OSM-3CA disposal requires its hyperactivity. Finally, we provide evidence that neurons also dispose of hyperactive kinesin-1 resulting from a clinic variant associated with amyotrophic lateral sclerosis, suggesting a widespread mechanism for regulating hyperactive kinesins.**

**Keywords** Hyperactive; Kinesin; Glia; Autoinhibition; Conformation
**Subject Categories** Cell Adhesion, Polarity & Cytoskeleton; Membranes & Trafficking; Neuroscience

## Introduction

The kinesin family of motor proteins plays a pivotal role in facilitating intracellular transport along microtubules (MTs) (Burute and Kapitein, 2019; Cason and Holzbaur, 2022; Christensen and Reck-Peterson, 2022; Hirokawa et al, 2009; Ou and Scholey, 2022). Comprising a motor domain responsible for ATPase-mediated conversion of chemical energy into mechanical forces, and a tail domain that interacts with specific cargo molecules or regulatory

proteins, kinesins exhibit stringent control over their catalytic activity (Burute and Kapitein, 2019; Ou and Scholey, 2022; Sweeney and Holzbaur, 2018). Cargo binding represents a key regulatory mechanism: In the absence of cargo loading, it is postulated that the kinesin tail undergoes a folding process, enabling interaction with the motor head and subsequent inhibition of ATPase activity. Conversely, cargo loading through the kinesin tail induces conformational changes that release autoinhibition and activate motors (Cai et al, 2007; Cason and Holzbaur, 2022; Coy et al, 1999; Espenel et al, 2013; Friedman and Vale, 1999; Guillaud et al, 2008; Hammond et al, 2009; Verhey and Hammond, 2009).

Dysregulation of kinesin disrupts the transport of vital molecules and organelles within neurons, contributing to neuronal dysfunction and degeneration (Sleigh et al, 2019). This impairment has been implicated in various neurodegenerative disorders, including Alzheimer's disease, Parkinson's disease, hereditary spastic paraplegia (HSP), and amyotrophic lateral sclerosis (ALS) (Akcimen et al, 2023; Blackstone et al, 2011; Feldman et al, 2022; Sleigh et al, 2019). While extensive investigations have focused on loss-of-function kinesin mutations, regarding the essential roles of various kinesins in extensive biological processes illustrated in knock-out mice (Hirokawa et al, 2009; Midorikawa et al, 2006; Nonaka et al, 1998; Tanaka et al, 1998; Zhao et al, 2001), recent discoveries have unveiled gain-of-function kinesin mutations associated with ALS and HSP (Akcimen et al, 2023; Anazawa et al, 2022; Baron et al, 2022; Brenner et al, 2018; Chiba et al, 2023; Chiba et al, 2019; Nakano et al, 2022; Nicolas et al, 2018; Pant et al, 2022). The precise mechanisms by which these mutations alleviate the kinesin autoinhibition and induce hyperactivity remain unclear. Existing kinesin inhibitors, although widely employed in biomedical researches, exhibit potent efficacy, limiting their suitability for therapeutic interventions (Rath and Kozielski, 2012). Thus, understanding the regula3tion of kinesin hyperactivation is essential for advancing disease treatment strategies.

Cilia represent a valuable in vivo model system for investigating protein conformational changes that drive motor activation or inactivation (Anvarian et al, 2019; Klena and Pigino, 2022; Nachury and Mick, 2019; Ou and Scholey, 2022; Reiter and Leroux, 2017). These microtubule-based organelles are widely distributed on the

[1]Tsinghua-Peking Center for Life Sciences, Tsinghua University, Beijing, China. [2]Beijing Frontier Research Center for Biological Structure, Tsinghua University, Beijing, China. [3]McGovern Institute for Brain Research, Tsinghua University, Beijing, China. [4]State Key Laboratory for Membrane Biology, Beijing, China. [5]School of Life Sciences, Tsinghua University, Beijing, China. [6]National Laboratory of Biomacromolecules, CAS Center for Excellence in Biomacromolecules, Institute of Biophysics, Chinese Academy of Sciences, 15 Datun Road, 100101 Beijing, China. [7]College of Life Sciences, University of Chinese Academy of Sciences, 100049 Beijing, China. [8]Physik Department E22, Technische Universitat Munchen, James-Franck-Strasse, Garching 85748, Germany. [9]These authors contributed equally: Chao Xie, Guanghan Chen. ✉E-mail: guangshuoou@tsinghua.edu.cn

surface of eukaryotic cells and play crucial roles in cell motility and sensory signaling (Anvarian et al, 2019; Nachury and Mick, 2019; Ou and Scholey, 2022). Defects in cilia function contribute to over 35 types of ciliopathies (Reiter and Leroux, 2017). The formation and maintenance of cilia rely on intraflagellar transport (IFT), a bidirectional movement along microtubules on the axoneme (Kozminski et al, 1993). During IFT, the motor proteins of MT-plus end-directed kinesin-2 family undergo a conformational change to activate at the ciliary base. Kinesin-2 then transports IFT particles, which are protein complexes loaded with ciliary precursors, to the tip of the axoneme (Anvarian et al, 2019; Klena and Pigino, 2022; Nachury and Mick, 2019; Ou and Scholey, 2022). Once the cargo is released, the kinesin-2 motor must undergo another conformational change to become inactivated, while the MT minus end-directed dynein-2 motor is activated to recycle the anterograde IFT machinery (Anvarian et al, 2019; Klena and Pigino, 2022; Nachury and Mick, 2019; Ou and Scholey, 2022).

In *Caenorhabditis elegans*, two members of the kinesin-2 family, heterotrimeric kinesin-II and homodimeric OSM-3 (osmotic avoidance abnormal of the nematode), cooperate to drive IFT and assemble sensory cilia in chemosensory neurons (Ou et al, 2005; Ou and Scholey, 2022; Prevo et al, 2015; Snow et al, 2004). Initially, the IFT particles are transported by kinesin-II and OSM-3 together to construct the middle ciliary segments. Subsequently, kinesin-II hands over the IFT particles to OSM-3, which then transports them alone to assemble the distal axoneme (Ou et al, 2005; Prevo et al, 2015; Snow et al, 2004). The cilium of *C. elegans*, characterized by its outward-facing microtubule plus ends, is an extension of the dendrite (Ishikawa and Marshall, 2017; Ou and Scholey, 2022). For both wild-type and mutant OSM-3 kinesin to reach the base of the cilium, they must transit the tens-of-micrometer-long dendrites, given their likely synthesis in the soma of sensory neurons. The transport of OSM-3 through the dendrite cannot be dependent on its own active transport as this would result in anterograde movement. It is plausible that OSM-3 either diffuses passively along the dendrite or behaves as inert cargo for the cytoplasmic dynein-1, which moves from the soma to the dendritic terminal (Ou and Scholey, 2022). In either scenario, it is imperative for OSM-3 to adopt an inactive conformation; otherwise, it would adhere to microtubules, impeding its transport or propelling it in an undesired direction.

Interestingly, a point mutation (G444E) in the predicted hinge region of OSM-3's coiled-coil stalk leads to abnormal activation of ATPase activity and robust processive movement of OSM-3 in vitro (Imanishi et al, 2006; Mohamed et al, 2018). However, the G444E allele specifically causes loss of the distal ciliary segments in *C. elegans*, exhibiting the same ciliary defects observed in *osm-3*-null animals (Starich et al, 1995; Yi et al, 2018). The reason why a kinesin that is hyperactive in vitro behaves as a loss-of-function motor in vivo has remained a long-standing mystery. Here, we present a proof-of-concept study that *C. elegans* neurons dispose hyperactive kinesins into glial cells for clearance.

## Results

### A hyperactive ciliary kinesin disrupted cilia

Using a single-molecule fluorescence assay, we have verified that the wild-type (WT) full-length OSM-3 kinesin does not exhibit processive movement along microtubules (MTs); however, the introduction of the G444E substitution in the hinge region of its coiled-coil stalk elicited robust processive movement (Figs. 1A,B and EV1A,B) (Imanishi et al, 2006; Mohamed et al, 2018). To monitor the conformational changes of the motor, we employed size exclusion chromatography coupled with multiple-angle light scattering (SEC-MALS) analysis to determine their molecular masses. Overlaying the elution profiles revealed that the G444E mutant protein exhibited a higher molecular weight than the WT protein (Fig. 1C,D), in agreement with the hydrodynamic properties observed through sucrose gradient sedimentation (Imanishi et al, 2006). These findings indicate that the substitution of G444E leads to the adoption of a more extended conformation by OSM-3, resulting in the generation of a constitutively active OSM-3 kinesin motor protein, which we henceforth refer to as OSM-3CA.

To unravel the mechanism by which OSM-3CA causes identical ciliary defects to *osm-3(p802)* null allele, we used a genome-editing technique to introduce the G444E mutation into a strain harboring the OSM-3::GFP (green fluorescence protein) knock-in (KI). All examined *osm-3::gfp* KI animals were capable of utilizing their cilia to uptake the fluorescence dye 1,1′-dioctadecyl-3,3,3′,3′-tetra-methylindocarbocyanine perchlorate (DiI) from the medium ($n > 100$), but none of the *osm-3CA::gfp* worms ($n > 100$) showed DiI uptake, indicating a compromised ciliary function or integrity, manifested as a Dyf phenotype. To visualize IFT and cilia, we introduced a red fluorescent protein, mScarlet, fused with the dynein-2 motor protein CHE-3, into *osm-3CA* animals. In contrast to the bidirectional motility of CHE-3::mScarlet observed in the full-length WT cilia, no fluorescence was detected in the distal ciliary segments of *osm-3CA::gfp* worms ($n > 50$), with CHE-3::mScarlet being restricted to the remaining middle ciliary segments, indicating the loss of distal ciliary segments (Fig. 1E,F). In addition, we performed transmission electron microscopy (TEM) on high-pressure frozen, freeze-substituted specimens of wild-type and *osm-3* mutant animals. Our examination of serial sections revealed that *osm-3CA* animals indeed lost their distal ciliary segments, resembling the defect observed in the osm-3 null allele (Fig. 1G). Furthermore, we observed ectopic insertions of singlet microtubules interspersed among the doublet microtubules in the remaining middle segments of the *osm-3(sa125)* strain, which expresses hyperactive OSM-3CA (Fig. 1G). This phenotype closely resembles that of the *osm-3(p802)* null allele, as detailed in our earlier study (Xie et al, 2020). We propose that the aberrant torque generation resulting from the absence of OSM-3 may disrupt the axonemal structure's nine-fold symmetric arrangement of doublet microtubules with the insertion of these ectopic singlets (Xie et al, 2020). Essentially, none of the *osm-3CA* heterozygotes displayed any ciliary defects (Fig. EV1D) ($n > 40$ heterozygotes examined), and the expression of WT *osm-3* DNA in ciliated neurons completely rescued the ciliary phenotypes in *osm-3CA* animals (Fig. 1H,I), indicating that *osm-3CA* functions as a recessive loss-of-function mutant.

We have recently reported that RNA editing restricts hyperactive ciliary kinases by modifying the kinase's own message RNAs, thereby reducing kinase translation (Li et al, 2021). Utilizing the established RNA editing analysis method, we did not detect any RNA editing events in the mRNA molecules of *osm-3* or *osm-3CA* (Fig. EV1G,H). Consequently, the discrepancy between the hyperactivity observed in vitro and the kinesin-null phenotype observed in vivo suggests the

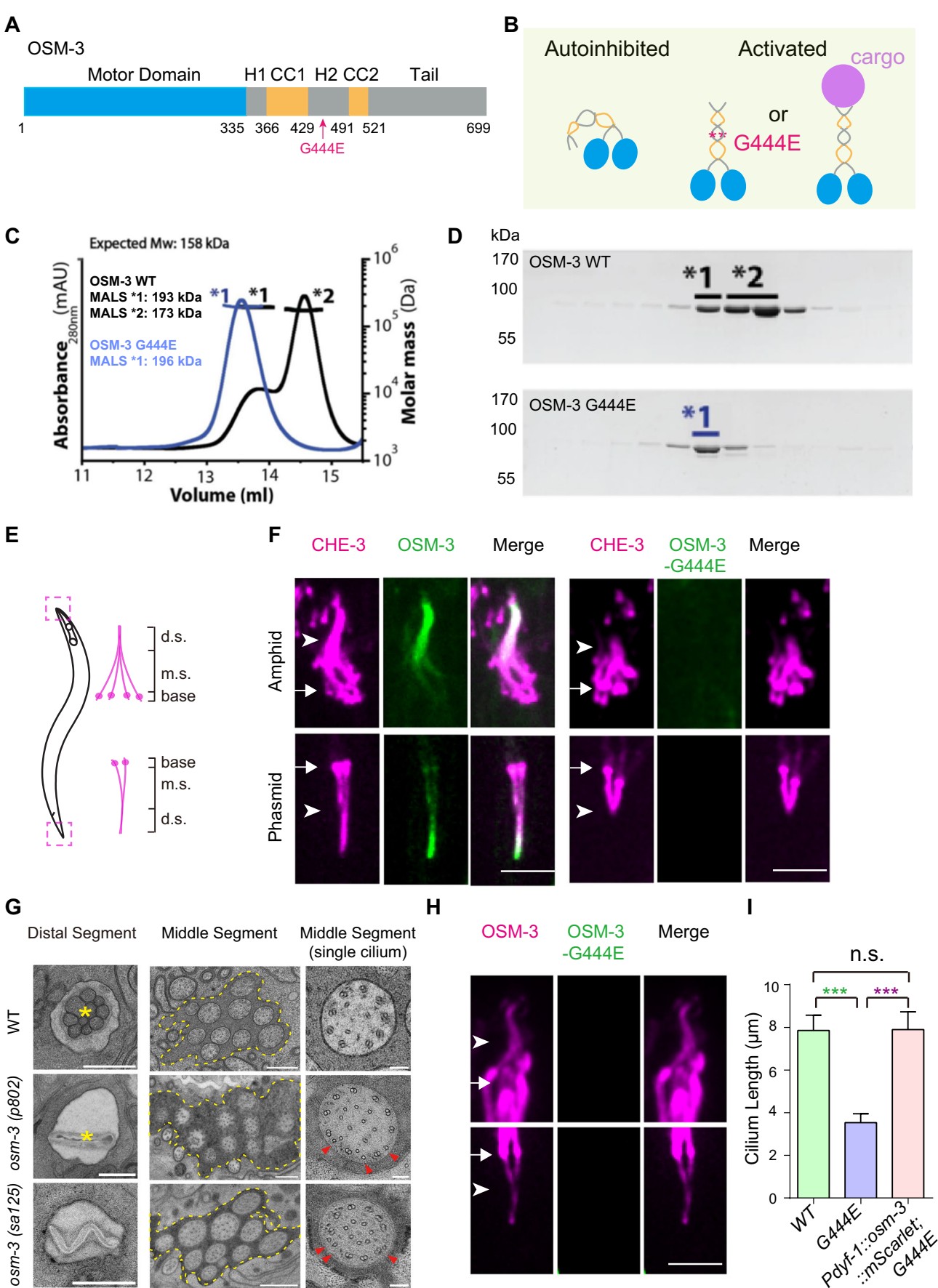

**Figure 1. A hyperactive OSM-3 kinesin disrupted cilia.**

(A) Domain architecture of the full-length OSM-3 kinesin motor protein. The motor domain (blue), coiled coils (yellow), hinge regions (h1 and h2) and tail domain (gray) are indicated. The residue numbers of each domain are shown below the bar graph. (B) An autoinhibition model of OSM-3 kinesin motor protein. Cargo attachment or the G444E mutation in h2 region converts the full-length OSM-3 from its intramolecular folded and autoinhibited conformation to an extended and active conformation that leads to an abnormal processive movement without cargo loading in vitro. (C) Overlay of the elution profiles of WT (line in black) and G444E mutant (line in blue) OSM-3. The left shoulder of the OSM-3 WT overlaps with the elution profile of the OSM-3 G444E. The molar masses determined from the expected mass and the MALS fit are shown on the left. (D) The SDS–PAGE analyses to identify the elution peaks shown in C. Protein constituents were determined by subsequent liquid chromatography-tandem mass spectrometry (LC-MS/MS) analysis. (E) Schematic of the *C. elegans* amphid and phasmid cilia. Each cilium contains a ciliary base, a middle segment (m.s.), and a distal segment (d.s.). The dashed boxes are enlarged on the right. (F) Ciliary localization of GFP-tagged wild-type (WT) OSM-3 or OSM-3-G444E protein in corresponding knock-in (KI) animals. Magenta shows mScarlet-tagged endogenous dynein-2 motor protein CHE-3. Arrows indicate the ciliary base. Arrowheads indicate the junctions between the middle and distal ciliary segments. Scale bars, 5 µm. (G) Representative transmission electron microscopy (TEM) images of the distal and middle ciliary segment from WT and *osm-3(p802, null)* or *osm-3(sa125, G444E)* mutant animals. The asterisks denote the presence of cilia in the WT amphid channel, while the cilia are absent in the *osm-3(p802, null)* or *osm-3(sa125, G444E)* alleles. Left and middle panels: overviews of the amphid channel. Right panels: the images of a single cilium. Scale bars, 500 nm (left and middle); 100 nm (right). Red arrowheads indicate ectopic singlet microtubules. (H) Representative fluorescence images of amphid and phasmid cilia. WT OSM-3 was expressed under the control of the ciliated sensory neuron-specific promoter P*dyf-1* in OSM-3-G444E::GFP KI animals. Arrows indicate the ciliary base. Arrowheads indicate the junctions between the middle and distal ciliary segments. Scale bar, 5 µm. (I) Quantification of cilium length in WT, *osm-3(G444E)* and the rescue strains. Green asterisk represent comparisons between WT and *osm-3(G444E)* animals; purple asterisk represents comparisons between *osm-3(G444E)* and the rescue strain. $N = 20$–$53$, number of worms analyzed. n.s. not significant; ***$P < 0.001$ by one-way ANOVA using BH method to adjust $P$ values. Data are mean ± SD. Source data are available online for this figure.

existence of a previously unknown mechanism that governs the regulation of OSM-3CA in a living organism.

## OSM-3CA is absent from cilia but forms puncta in axons or outside of neurons

The endogenous OSM-3::GFP protein predominantly localizes and moves within the distal segments of the cilia (Fig. 1F) (Prevo et al, 2015; Xie et al, 2020). However, upon examining the fluorescence of OSM-3CA::GFP, a remarkable finding emerged: OSM-3CA does not localize to the remaining middle ciliary segments, nor does it exhibit signal along the dendrites (Fig. 1F). Instead, the OSM-3CA protein forms puncta in the region located anterior to the soma (Fig. 2A), which are not found in the non-fluorescently tagged *osm-3(sa125)* strain, eliminating the possibility of nonspecific autofluorescence (Fig. EV1C). The number of OSM-3CA puncta increases from the L1 to L3 larval stage but keeps constant from L3 to adult animals (Fig. 2B). To visualize the nuclei and membrane of sensory neurons, we expressed TagBFP-tagged histone H2B and mScarlet-tagged myristoylation sequence under the control of the ciliated neuron-specific promoter P*dyf-1* (Ou et al, 2005) in *osm-3CA::gfp* animals (Fig. 2C). We confirmed that the OSM-3CA signal is absent from the cilia or dendrites but found that OSM-3CA localizes around axons (Fig. 2C). Notably, a substantial proportion of OSM-3CA puncta do not overlap with the red fluorescence that marks the neuron periphery, indicating their presence outside of the boundaries of the neurons themselves (Fig. 2C,D). In the OSM-3::GFP/OSM-3CA::GFP heterozygotes, GFP fluorescence localizes within cilia and also forms puncta (Fig. EV1D,E). In *osm-3CA::gfp* animals, overexpressed OSM-3WT::mScarlet localizes in cilia, whereas OSM-3CA::GFP still forms puncta outside cilia (Figs. 1H and EV1F). Thus, OSM-3CA protein may not interfere with the wild-type OSM-3 protein in the same neurons, which aligns with its recessive and loss-of-function nature.

## OSM-3CA is disposed through membrane abscission at the tips of neurites

How can a cytosolic kinesin be present outside the cells that produce it? In *osm-3CA::gfp* animals, we observed abnormal

branching of neurites from the stereotypical axons of sensory neurons (Figs. 2E and EV2A,B). We found that OSM-3CA::GFP puncta accumulated at the tips of the ectopic neurites (Figs. 2E,F and 2A,B). Our fluorescence time-lapse microscopy documented the pinching off processes of OSM-3CA::GFP puncta from the neurite tips, forming balloon-like extrusions that subsequently moved away from the neurites (Figs. 2G and EV2C,D; Movies EV1–EV3). The released puncta displayed a plasma membrane marker from the P*dyf-1::myristoylation::mScarlet* reporter, indicating their association with membranes (Figs. 2F,G and EV2C,D; Movies EV1–EV3).

To better elucidate the production mechanism of OSM-3CA puncta, we examined the actin and microtubule cytoskeletons and the ESCRT complex responsible for membrane abscission (Vietri et al, 2020), during OSM-3CA puncta production. In pursuit of this, we constructed a mScarlet knock-in strain that marks OSM-3CA with the red fluorescence, subsequently establishing GFP reporter lines to visualize the cytoskeleton and ESCRT. We discerned a ring-like configuration of the microtubule cytoskeleton within some of OSM-3CA puncta (Fig. EV3H). In addition, the OSM-3CA fluorescence in the puncta manifested as filamentous assemblies exhibiting a rotational dynamism (Movie EV4, $n = 5$), which suggests that OSM-3CA might navigate the annular microtubule scaffolding, engendering the rotational behavior within the puncta.

We employed the GFP-tagged actin-binding domain of the moesin protein to investigate the actin cytoskeleton and utilized TSG-101 to track the ESCRT complex (Chai et al, 2012; Vietri et al, 2020). Relative to their fluorescence distribution along the neurites, there was a pronounced enrichment of GFP fluorescence associated with both the actin cytoskeleton and the ESCRT marker within OSM-3CA puncta (Fig. EV3A–G; Movies EV5–6). These findings suggest a potential involvement of the actin cytoskeleton and the ESCRT machinery in the genesis of OSM-CA puncta, establishing a foundation for subsequent investigations utilizing mutations compromised in actin and ESCRT functionality. Furthermore, the future study will examine the contribution of the ESCRT pathway when an effective and specific pharmacological inhibitor targeting ESCRT is available.

The measured diameter of the OSM-3CA::GFP puncta was ~1.4 µm, exceeding the size range of typical exosomes (30-100 nm

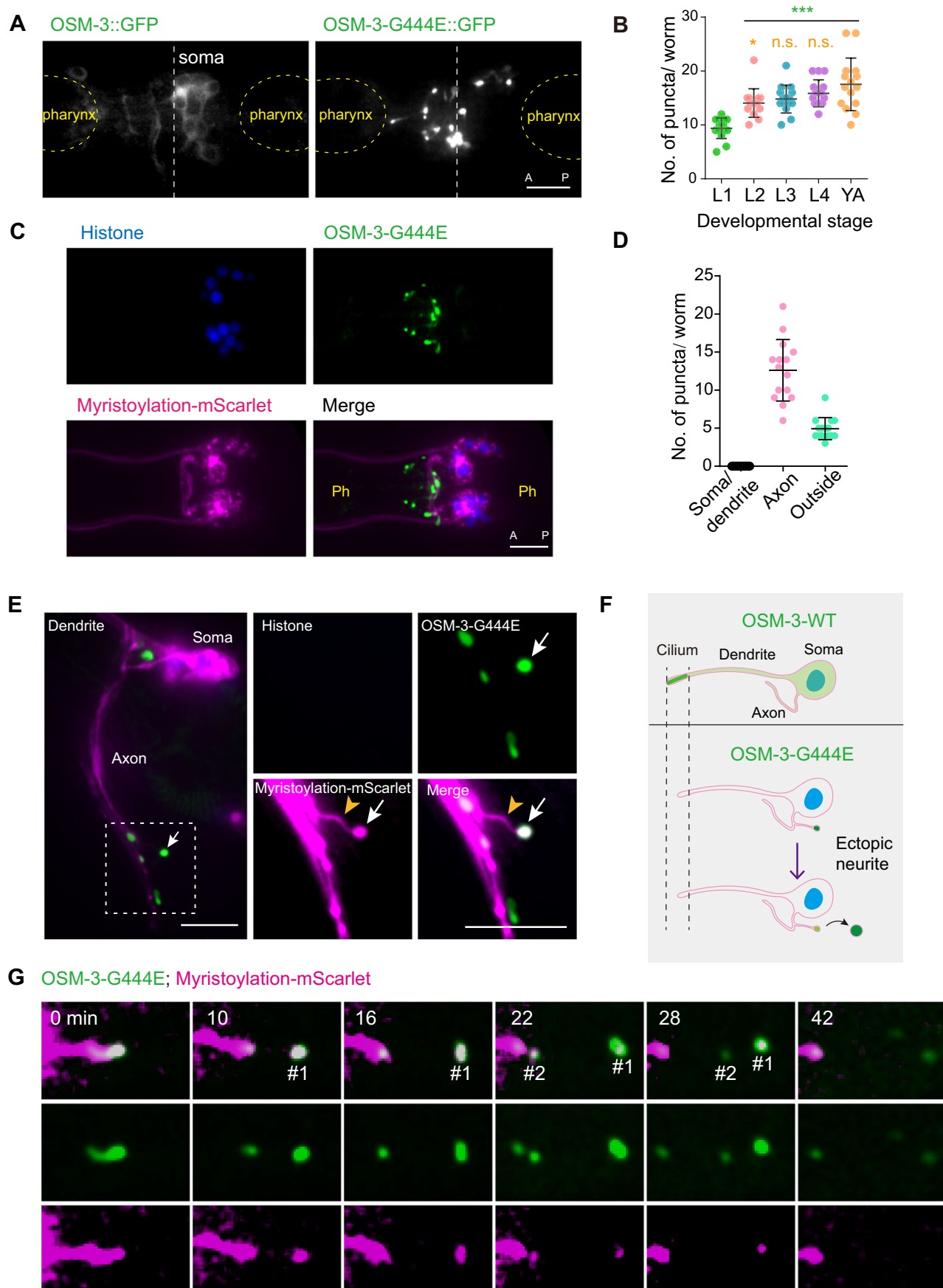

**Figure 2.  OSM-3CA is absent from cilia but forms granules in axons or disposed through membrane abscission at the tips of neurites as ejecta.**

(A) Representative fluorescence images of the endogenous OSM-3-G444E that abnormally accumulate between two pharynxes compared to WT OSM-3. The white dotted lines indicate the center between the two pharynxes. Scale bar, 10 μm. A anterior, P posterior. (B) Statistics of the number of OSM-3-G444E puncta at different developmental stages. YA young adult. $N = 15$ animals for each stage. Green asterisks represent comparisons between L1-stage and the other four-stage animals; orange asterisks represent comparisons between L2–L4-stage and young-adult animals. n.s. not significant; *$P < 0.05$, ***$P < 0.001$ by one-way ANOVA using BH method to adjust $P$ values. Data are mean ± SD. (C) Representative fluorescence images of OSM-3-G444E localization relative to the ciliated sensory neurons. The histone marker HIS-54::BFP and membrane marker Myristoylation::mScarlet are expressed under the control of P*dyf-1* (a ciliated neuron-specific promoter) in OSM-3-G444E::GFP KI animals. Ph pharynx. Scale bar, 10 μm. A anterior, P posterior. (D) Statistics of the distribution of OSM-3-G444E puncta at the young adult stage. $N \geq 15$ animals. Data are mean ± SD. (E) Representative fluorescence image of OSM-3-G444E puncta at the tips of axons or ectopic neurites. The histone marker HIS-54::BFP and the membrane marker Myristoylation::mScarlet are expressed under the control of the ciliated sensory neuron-specific promoter P*dyf-1* in OSM-3-G444E::GFP KI animals. The dashed box is enlarged on the right. Orange arrowheads, an ectopic neurite; arrows, the tip of the neurite. Scale bars, 10 μm. (F) Schematic diagram of the fate of hyperactive OSM-3 in sensory neurons. (G) Fluorescence time-lapse images of endogenous OSM-3-G444E in the sensory neuron. #1 and #2 indicate the two sequentially released OSM-3-G444E puncta. Scale bar, 5 μm. Source data are available online for this figure.

in diameter) and other microvesicles (several hundred nanometers in diameter), but smaller than typical exophers, which have a diameter of 3.8 μm. (Melentijevic et al, 2017; Pegtel and Gould, 2019; van Niel et al, 2018). The particle size may appear to be a subtle distinction, and extracellular vesicles are remarkably diverse, regarding their different biogenesis and biological functions (Buzas, 2022). Large exophers, identified recently (Melentijevic et al, 2017), are extruded from neuronal cell bodies, appear to function in "trash" elimination, and do not appear to share cytoskeletal features or biogenesis machinery (marked by TBB-2, actin, and ESCRT) (Fig. EV3) with OSM-3CA extraneuronal puncta and yet have not been molecularly characterized, leaving their precise biomarkers unclear. In addition, the process governing the release of exophers from the cell body remains an enigmatic aspect of cell biology, and to date, no pharmacological agent has been identified that can effectively disrupt or alter this exopher release mechanism. Given the lack of characterization of the contents and ultrastructural analysis of puncta associated with OSM-3CA, we henceforth refer to the ejected OSM-CA puncta as OSM-3CA ejecta.

## Dynamics of OSM-3CA within ejecta and neurites

Our fluorescence recovery after photobleaching (FRAP) experiments shed light on the dynamic nature of OSM-3CA within ejecta. When we photobleached a segment of the OSM-3CA ejecta, characterized by rotational movement, we observed a remarkably rapid recovery half-time of just 0.26 s, with the recovery level reaching 47.3% (Fig. 3A,B). This finding emphasizes that OSM-3CA ejecta are not static protein aggregates but rather dynamic structures. This dynamic recovery is likely facilitated by the movement of OSM-3CA along microtubules or its diffusion from unbleached regions, as suggested by the filamentous TBB-2 signals within the OSM-3CA ejecta and its filament-like movements (Movie EV4; Fig. EV3H).

To delve deeper into the process of OSM-3CA incorporation into ejecta from neurites, we conducted full-area bleaching on OSM-3CA ejecta connected to such neurites. Tracking fluorescence changes in both the ejecta and neurites revealed a significantly lengthened recovery half-time of 5.0 s in the ejecta, in stark contrast to the 0.26 s observed in partial bleach experiments (Fig. 3C–E). This extended recovery period mirrors the fluorescence decay in the neurites, indicating a continuous and rapid accumulation of hyperactive OSM-3CA in the ejecta.

However, when we subjected the entire OSM-3CA ejecta detached from the neurites to bleaching, we observed a minimal fluorescence recovery of only 3.9% (Fig. 3F,G). This limited recovery is consistent with the expectation that no new OSM-3CA protein influx would occur after the ejecta's detachment from the neuron, where OSM-3CA is originally produced.

## The disposed OSM-3CA were engulfed by neighboring glial cells

What happens to the disposed OSM-3CA ejecta? We observed a continuous disposal of OSM-3CA ejecta by sensory neurons, but the number of OSM-3CA ejecta keep constant after L3 larval stage (Fig. 2B). Moreover, the OSM-3CA ejecta gradually disappeared after being disposed (Figs. 2G and EV2C,D; Movies EV1–3). These data suggest the possibility of internal degradation of OSM-3CA by the engulfing cells. The *C. elegans* amphid sensory organ consists of ciliated sensory neurons and glial supporting cells, and axons of bilateral amphid entering the nerve ring around the pharynxes are arranged in bundles and enveloped by CEPsh glia. (Fig. 4A) (Singhvi and Shaham, 2019; Ward et al, 1975). We investigated whether the glial cells play a role in engulfing and reducing the OSM-3CA ejecta released from sensory neurons. By utilizing promoter P*hlh-17* to specifically drive the expression of mScarlet in the cephalic (CEP) sheath glia (Singhvi and Shaham, 2019), we detected a significant colocalization of OSM-3CA ejecta with the red fluorescence (Fig. 4B), which supports the hypothesis that glial cells contribute to OSM-3CA clearance.

The engulfment receptor CED-1 plays a critical role in the degradation of various cellular remnants, including apoptotic cell corpses, cytokinetic midbodies, and exophers (Chai et al, 2012; Melentijevic et al, 2017; Zhou et al, 2001). The introduction of the *ced-1* null allele into OSM-3CA animals resulted in a significant increase in the number of OSM-3CA ejecta, indicating the involvement of the CED-1 receptor in OSM-3CA clearance (Fig. 4C,D). This effect could be reversed by expressing the *ced-1* receptor under its own promoter (Fig. 4C,D). Furthermore, when *ced-1* was expressed under the control of the glial cell-specific promoter P*hlh-17* in *ced-1* mutant animals, there was a marked reduction in the abnormally elevated number of OSM-3CA ejecta, returning them to a level comparable to that observed in wild-type animals (Fig. 4C,D). In this strain, the size and fluorescence intensity of OSM-3CA ejecta were even lower than those in wild-type animals (Fig. 4C,D). This suggests that augmented *ced-1* expression in glia might bolster their phagocytic and subsequent degradative capacities, thereby mitigating OSM-3CA levels.

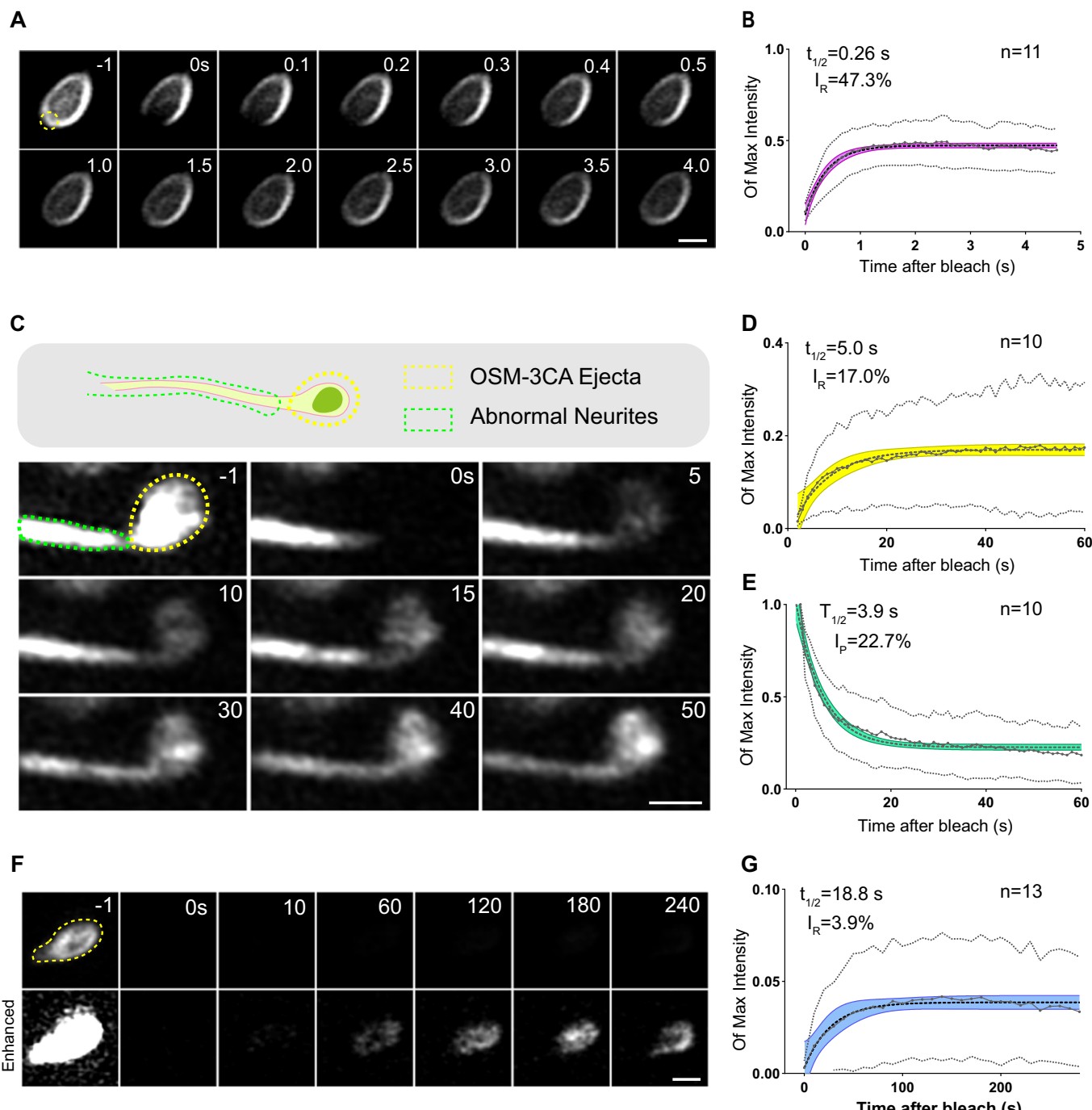

**Figure 3. Dynamics of OSM-3CA within ejecta and neurites.**

(**A**) Representative FRAP images of released OSM-3CA ejecta after partial photobleaching. Endogenous GFP-tagged OSM-3CA ejecta are partially bleached and images are taken every 100 ms. Yellow dashed line marks the bleached region. Scale bar, 1 μm. (**B**) Statistics of fluorescence recovery of bleached regions in (**A**). Gray dots with lines are mean values of each time point while dotted lines represent SD. The dashed line is the recovery curve fitted with an exponential equation as described in "Methods". Colored full lines show 95% CI (confidence interval) of the fitted curve. $t_{1/2}$, time used to achieve half of the maximum recovered intensity. $I_R$ maximum recovered intensity, $n$ total number of events. (**C**) upper panel, schematics of the photobleaching experiment. Unreleased OSM-3CA ejecta are marked with yellow dashed lines while abnormal neurites are marked with green dashed lines. Yellow regions are photobleached; Lower panel, representative FRAP images after photobleaching of the ejecta regions. Images are taken every second. Scale bar, 2 μm. (**D**) Statistics of fluorescence recovery of yellow region marked ejecta that are photobleached in (**B**). (**E**) Statistics of fluorescence intensity in abnormal neurites in (**B**) after photobleaching. Curve is fitted with an exponential equation as described in "Methods". $T_{1/2}$ fluorescence half-life, $I_P$ plateau value indicates the minimum intensity in the region after it is balanced. (**F**) Representative FRAP images of released OSM-3CA ejecta after full-region photobleaching. Lower panel shows the enhanced images of the upper panel. Images are taken every 10 s. Scale bar, 1 μm. (**G**) Statistics of fluorescence recovery of yellow region marked ejecta that are photobleached in (**F**). Source data are available online for this figure.

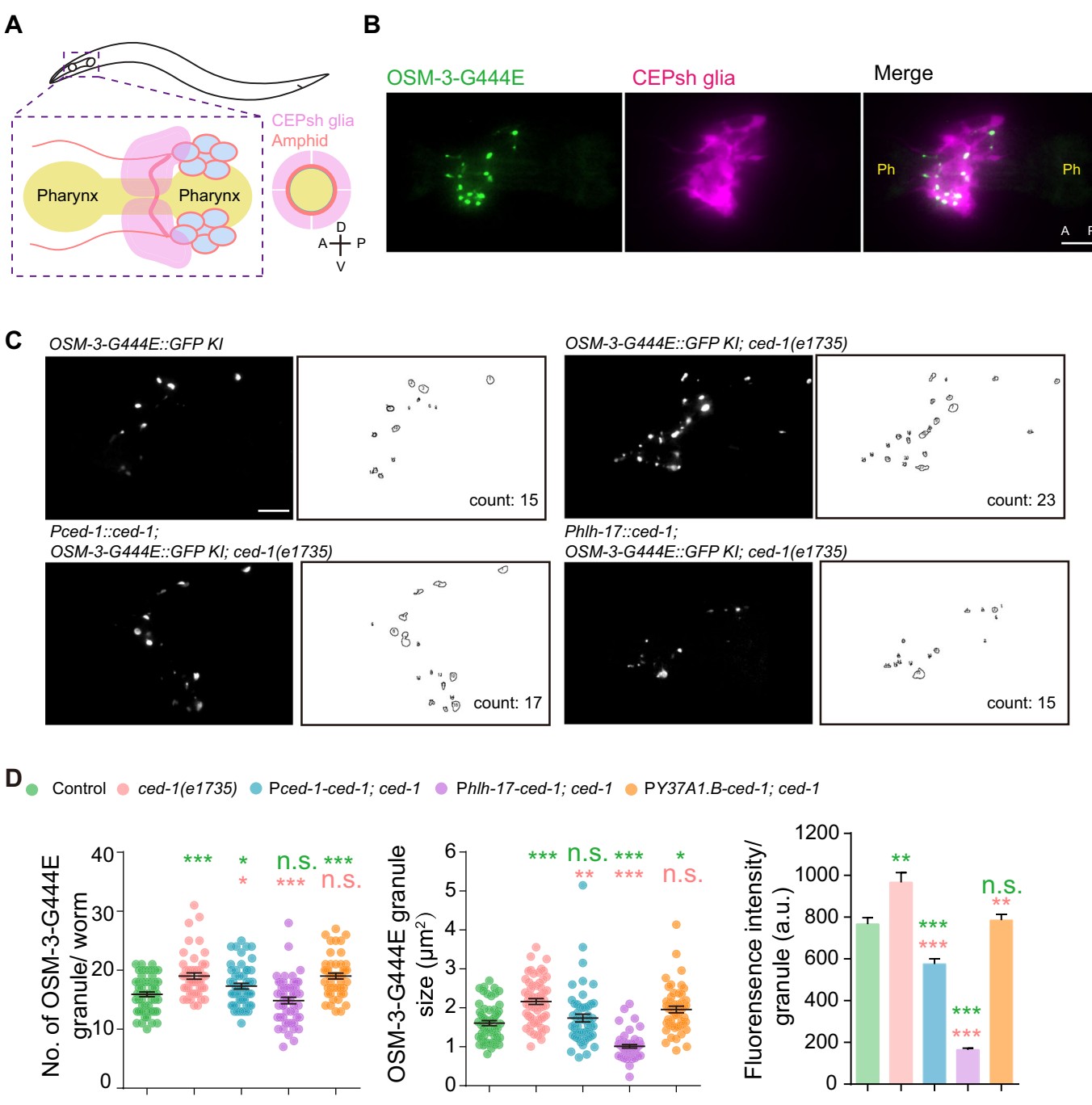

**Figure 4. The disposed OSM-3CA were engulfed by neighboring glial cells.**

(A) Schematic depiction of the relative position of postembryonic amphid neurons and CEPsh glia. Axons of bilateral amphid entering the nerve ring around the pharynx are arranged in bundles and enveloped by CEPsh glia. A anterior, P posterior, D dorsal, V ventral. (B) Fluorescence images of endogenous OSM-3-G444E and CEPsh glia. mScarlet is expressed under the control of a CEPsh glia-specific promoter P*hlh-17* in OSM-3-G444E::GFP KI animals. Ph pharynx. Scale bar, 10 μm. A anterior, P posterior. (C) Representative fluorescence images (left) and quantification scheme (right) of OSM-3-G444E puncta in the animals of the indicated genotypes. Count, the number of puncta. Scale bar, 10 μm. (D) Quantification of the number, size and fluorescence intensity of OSM-3-G444E puncta in the animals of different genotypes. $N \geq 50$ animals for all genotypes (left and middle). Each dot represents the average measurements of 5–35 individual puncta in one worm (middle). $N = 10$ animals, 168–244 puncta for per strain (right). Green asterisks represent comparisons between OSM-3-G444E::GFP KI animals (Control) and other strains; pink asterisks represent comparisons between *OSM-3-G444E::GFP; ced-1(e1735)* double mutants and the rescue strains. n.s. not significant; *$P < 0.05$, **$P < 0.01$, ***$P < 0.001$ by one-way ANOVA using BH method to adjust $P$ values. Data are mean ± SEM. Source data are available online for this figure.

Consistently, Chiu et al, demonstrated that expression of *ced-1* in muscle-based engulfing cells effectively mitigates the retention of axon debris observed in *ced-1* mutants (Chiu et al, 2018). Notably, when *ced-1* was overexpressed, the accumulation of axon debris was reduced to levels even lower than those observed in wild-type animals (Chiu et al, 2018), which suggests that *ced-1* overexpression in engulfing cells may also enhance their capacity for extracellular debris clearance. On the other hand, when the *ced-1* gene was introduced into the hypodermis of *ced-1* mutant animals using the hypodermis-specific promoter P*Y37A1.B* (Yang et al, 2017), there was no noticeable reduction in the number of OSM-3CA ejecta (Figs. 4C,D and EV4A,B). This suggests that the clearance of OSM-3CA ejecta depends on the activity of the CED-1 receptor, which specifically operates from glial cells surrounding sensory neurons.

Considering there are extensive pathways involved in the engulfment of cell corpses or debris (Conradt et al, 2016), we further examined the potential role of the parallel pathways in OSM-3CA ejecta removal. We observed no notable augmentation in the count of OSM-3CA ejecta in either *ced-2* or *ced-5* mutant animals (Fig. EV4C,D). The generation of the *ced-1; ced-2* double mutant did not manifest an enhanced ejecta presence relative to the *ced-1* single mutants (Fig. EV4E,F). These findings indicate that the pathways governed by *ced-2* and *ced-5* are dispensable for the clearance of ejecta.

## Genetic suppressor screens identified intragenic suppressors of OSM-3CA

We conducted a forward genetic suppressor screen to search for mutations that could restore the ciliary function of *osm-3CA* (Fig. EV5A). Following an extensive screen of over $10^5$ haploid genomes, we successfully isolated and cloned 22 suppressor mutants demonstrating DiI uptake comparable to that of wild-type organisms (Fig. EV5B) (n = 100 for each). Among these mutants, 13 of them were mapped to be intragenic suppressors, leading to six missense mutations within the motor domain of OSM-3 (Figs. 5A,B and EV5B,C). Eight independent alleles carry the same intragenic R238W mutation within OSM-3 (Fig. EV5B). To validate the suppressive effects of R238W and H207Q, we introduced the lesion of *osm-3(R238W)* or *osm-3(H207Q)* into OSM-3::GFP animals by genome editing. Each mutation exhibited profound impairments in dye filling, ciliary length, and IFT velocity (Fig. 5C–E), consistent with their pivotal roles in the motor domain. Subsequently, we generated strains with double knock-in mutations of *osm-3(R238W;G444E)* and *osm-3(H207Q;G444E)*. The resultant mutant animals mirror the observations made in the double mutants acquired through genetic suppressor screens (Figs. 5C and EV5D). Through another genetic suppressor screen that aimed to restore ciliary defects of the *osm-3H207Q* single mutant, we isolated three independent alleles carrying G445E as the suppressor (Fig. EV5D). G444E and G445E mutations occur at the same hinge region and may have similar impacts on OSM-3. These results indicate that R238W and H207Q represent intragenic mutations that rectify the ciliary defects induced by the OSM-3G444E mutation.

By what mechanism does a second missense substitution within the kinesin motor domain ameliorate the defects induced by the initial mutation in the hinge region? The ATP-binding motifs within the OSM-3 motor domain, akin to other extensively studied kinesin motors, encompass the P-loop, the phosphate-sensing loop

known as Switch 1, and Switch 2 (Sweeney and Holzbaur, 2018; Varela et al, 2021). H207Q is situated in Switch 1 while R238W adjoins Switch 2 (Varela et al, 2021), with each substitution likely impeding nucleotide binding, thereby diminishing ATPase activity. Through microtubule-stimulated ATPase activity assays, we established that H207Q and R238W markedly attenuated ATPase activity to 18% and 39% comparing to the wild-type OSM-3, respectively (Fig. 5F). Furthermore, H207Q; G444E and R238W; G444E double mutants reduced the hyperactivity exhibited by OSM-3G444E which is over three times higher than that of wild-type OSM-3, to a level even lower than the wild-type but higher than that conferred by each individual mutation (Fig. 5F).

To examine the processivity of OSM-3, we used a total internal reflection fluorescence (TIRF) microscope to visualize GFP-labeled motors. Our findings unveiled that either H207Q or R238W transformed the processive OSM-3G444E motor into a WT OSM-3 counterpart, which exhibited no processivity in single-molecule assays (Fig. 5G). Furthermore, in our microtubule gliding assays, we observed that OSM-3R238W demonstrated microtubule gliding at a velocity twice as rapid as the WT, on par with the speed of OSM-3G444E (Figs. 5G and EV5E; Movie EV7). Notably, the OSM-3R238WG444E double-mutant protein propelled MTs marginally faster than OSM-3G444E alone, indicating that R238W does not impede force generation of the motor protein (Figs. 5G and EV5E; Movie EV7). In contrast, no MT-gliding activity was discernible in the OSM-3 proteins harboring H207Q (Fig. 5G), indicating that H207Q disrupts force generation. These results show that H207 and R238W potentially rescue the effects of G444E through distinct mechanisms.

Next, we studied the in vivo behaviors of the single and double mutants of OSM-3. We observed that the vast majority of GFP fluorescence was localized within cilia of the *osm-3(H207Q;G444E)* and *osm-3(R238W;G444E)* double-mutant animals (Fig. 5C). Notably, neither of the double mutants displayed any discernible OSM-3CA::GFP ejecta throughout the entirety of the *C. elegans* body (N > 100 for each double mutant), revealing the complete elimination of OSM-3CA ejecta (EV Fig. 5F). In addition, we noted an absence of OSM-3CA::GFP ejecta in all four other examined intragenic suppressor strains as well (N > 50 for each strain).

OSM-3 carrying the H207Q mutation displayed kinesin-II-mediated slow motility exclusively within the middle segments of the cilia, while failing to enter or construct the distal segments (Fig. 5C,E). This observation is consistent with the absence of detectable MT-gliding activity exhibited by OSM-3H207Q and implies that OSM-3H207Q may serve as a non-functional kinesin motor (Fig. 5G). Conversely, the OSM-3R238W single mutant, which demonstrated MT gliding in vitro, exhibited partial construction of the distal ciliary segments and moved along them (Figs. 5C,E,G and EV5E), indicating that OSM-3R238W retains partial motor functionality in vivo. Remarkably, both the OSM-3H207QG444E and OSM-3R238WG444E double mutants moved along both the middle and distal ciliary segments (Figs. 5C,E and EV5D). Although the IFT speeds of both double mutants were slower compared to those of WT OSM-3, the double-mutant animals generated slightly elongated cilia in comparison to WT worms (Fig. 5D,E). Collectively, these in vivo findings are consistent with biochemical results, indicating that the H207Q and R238W mutations curtailed the hyperactivity of OSM-3G444E

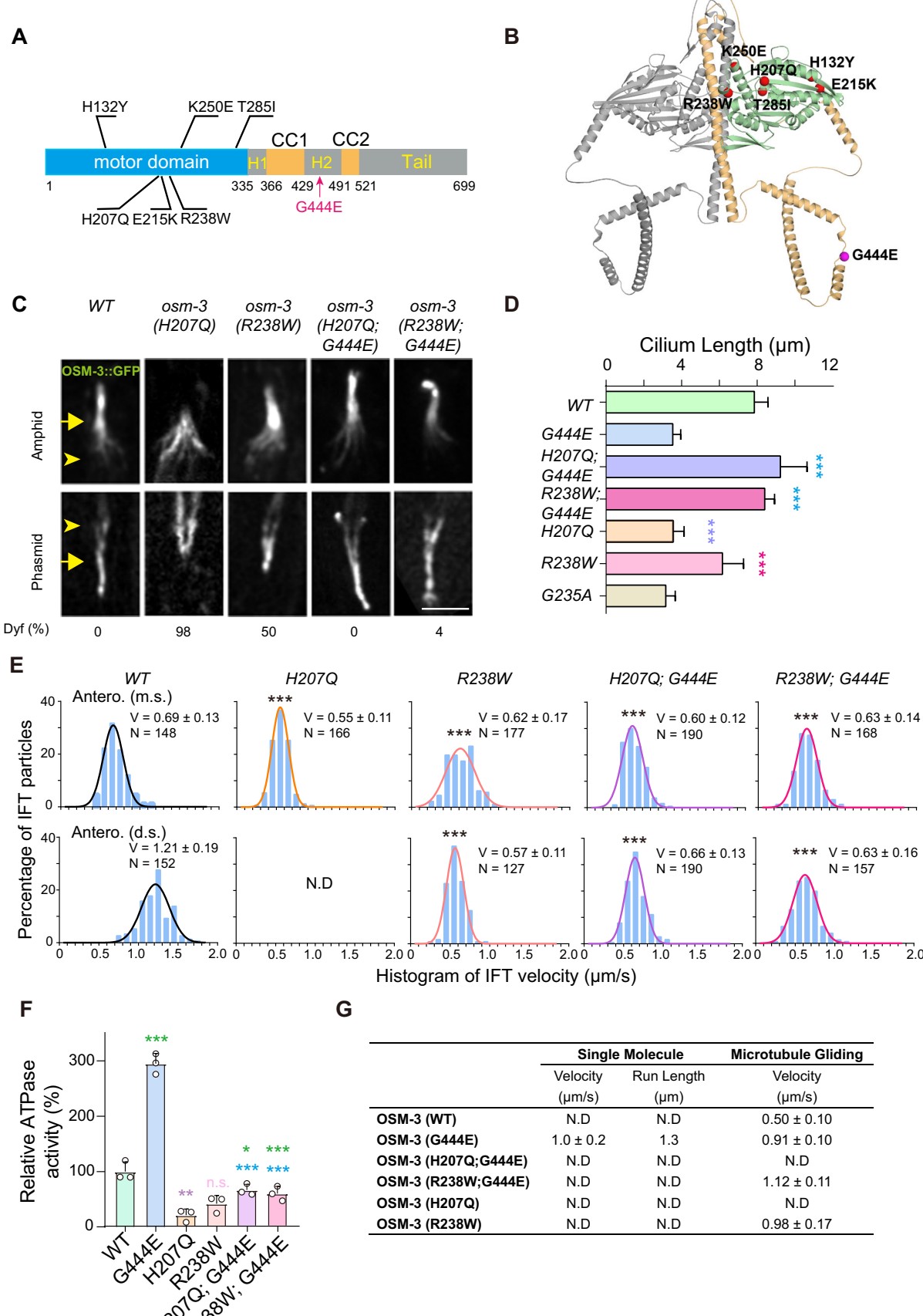

◄

**Figure 5. Intragenic mutations of *osm-3* restore the ciliary localization of OSM-3-G444E and rescue ciliary defects in *osm-3(G444E)*.**

(A) Schematic of the mutations on OSM-3 that suppress the phenotypes of *osm-3(G444E)*. (B) The intragenic mutations on OSM-3. Structure of dimerized OSM-3 was predicted by Alphafold2 using LocalColabFold. Motor domain is labeled with green in one copy of OSM-3 and the stalk and tail regions are labeled with wheat color. (C) Amphid and phasmid cilia in the WT, H207Q mutant *osm-3(H207Q)*, R238W mutant *osm-3(R238W)* animals, and the two suppressors *osm-3(H207Q; G444E)*, *osm-3(R238W; G444E)*. Cilia are visualized with GFP-tagged endogenous WT OSM-3 or the corresponding mutant OSM-3. Dyf, dye-filling defective; $N \geq 100$. Arrowheads indicate the ciliary base. Arrows indicate the junctions between the middle and distal segments. Scale bar, 5 µm. (D) Quantification of cilium length in WT and *osm-3* mutant animals. Statistical significance compared to the control with matching color codes. $N = 27–53$, number of worms analyzed. ***$P < 0.001$ by one-way ANOVA using BH method to adjust $P$ values. Data are mean ± SD. (E) Histogram of anterograde IFT speeds of WT and mutant OSM-3 motor at the middle (top) and distal (bottom) ciliary segments. Velocities (V) and numbers of IFT particles (N) are indicated. The plots were fit by a Gaussian distribution. Comparisons were performed between the WT and the mutants. ***$P < 0.001$ by unpaired Student's $t$ tests. Data are mean ± SD. (F) Relative ATPase activity of WT and mutant OSM-3. The assays were performed as described in methods following the established protocols. The mean activity of WT OSM-3 was set to 100%. The results were obtained from three independent experiments. Statistical significance compared to the control with matching color codes. *$P < 0.05$, **$P < 0.01$, ***$P < 0.001$ by one-way ANOVA using BH method to adjust $P$ values. Data are mean ± SD. (G) Motility properties of WT and mutant OSM-3. N.D. not detected. Data are mean ± SD. Source data are available online for this figure.

and effectively rescued the resulting ciliary defects while eliminating ectopic OSM-3CA ejecta.

## Fluorescence lifetime imaging microscopy (FLIM) of OSM-3 and mutants

Next, we studied the conformation of OSM-3 and its mutants in vivo. As indicated by biochemical findings, OSM-3, akin to its mammalian counterpart KIF17, adopts a compact conformation in the autoinhibited state, whereas an extended conformation reflects their active state (Espenel et al, 2013; Hammond et al, 2010; Imanishi et al, 2006). Employing fluorescence lifetime imaging (FLIM) to assess intramolecular Förster resonance energy transfer (FRET) efficiency, a previous study devised an effective biosensor for KIF17 to monitor its autoinhibited and active conformations within living cells (Espenel et al, 2013). In this study, the motor protein was labeled with red and green fluorescence proteins at the N- and C-termini, respectively. Consequently, autoinhibited motors, adopting their compact conformation, engender FRET, leading to a reduction in the lifetime of GFP due to quenching by the red fluorescence protein. Conversely, active motors, in their extended conformation, do not exhibit such effects (Espenel et al, 2013). FRET-based methodologies have also proven successful in discerning populations of active and inactive kinesin-1 or kinesin-3 motors (Cai et al, 2007; Hammond et al, 2009) as well as investigating kinesin-cargo interactions (Guillaud et al, 2008). Following the same design principle, we aimed to establish the FLIM assay to distinguish the extended, active conformation from the compact, inactive state of OSM-3 in sensory neurons of *C. elegans*.

As a demonstration of the FLIM technique in *C. elegans*, we generated a transgenic strain expressing tandem GFP and mScarlet proteins at the C-terminus of OSM-3 (Fig. 6A). We observed a GFP lifetime of 2.62 ± 0.02 ns from the OSM-3::GFP::mScarlet fusion protein in the soma, which exhibited a significantly reduced lifetime compared to the non-FRET control of OSM-3::GFP ($\tau = 2.86 \pm 0.02$ ns) measured using the identical approach (Fig. 6B,C). Subsequently, we expressed mScarlet::OSM-3::GFP in *osm-3(p802)* null animals and successfully rescued its ciliary defects (100% fluorescence DiI uptake, $N > 50$ animals), demonstrating that the N- and C-terminal tags did not interfere with the OSM-3 functionality. The mScarlet::OSM-3::GFP construct exhibited a GFP lifetime of 2.62 ± 0.02 ns, similar to the positive FRET control of OSM-3::GFP::mScarlet (Fig. 6B,C), supporting the close proximity between N- and C-termini of OSM-3 in the soma of

sensory neurons. In line with the FRET-FLIM observations in the soma, a reduction in GFP lifetime was discerned in the dendrite for the mScarlet::OSM-3::GFP construct, hinting at the autoinhibitory nature of the wild-type OSM-3 in this location (Fig. 6F,G).

After validating the assay, we evaluated the GFP lifetime in OSM-3 mutants. Specifically, mScarlet::OSM-3G444E::GFP displayed a GFP lifetime of 2.87 ± 0.03 ns, which closely resembled the non-FRET control of OSM-3::GFP ($\tau = 2.86 \pm 0.02$ ns), indicating the absence of FRET between the tail and motor head of OSM-3G444E (Fig. 6D,E). Together with in vitro single-molecule findings and SEC-MALS results of OSM-3G444E (Figs. 1C,D and EV1A,B), the lifetime alteration caused by the G444E mutation provides in vivo evidence that this hinge mutant transforms the OSM-3 motor into an extended conformation.

Upon introducing the H207Q or R238W mutation into mScarlet::OSM-3G444E::GFP, we observed a reversal in the GFP lifetime to 2.72 ± 0.02 ns and 2.75 ± 0.02 ns, respectively (Fig. 6D,E). This finding suggests that the suppression of OSM-3G444E by these mutations may arise from the restoration of the extended conformation of OSM-3G444E to a folded, wild-type-like conformation. Although we were unable to obtain a sufficient quantity of OSM-3H207QG444E, or OSM-3R238WG444E double-mutant proteins for SEC-MALS analyses, the individual OSM-3H207Q or OSM-3R238W single mutant proteins exhibited a more compacted conformation compared to OSM-3G444E in vitro (Fig. EV5G). Therefore, in addition to the reduction of the ATPase activity (Fig. 5F), H207Q or R238W perhaps rescued the hyperactivity of OSM-3G444E by reinstating its compact conformation. Although the precise structural mechanisms underlying this conformational rescue effect remain unclear, the intragenic mutations may affect the overall conformation and dynamics of OSM-3, leading to a reduction in its hyperactive behavior.

## Mutations of the dyf-5 kinase were identified as intergenic suppressors of OSM-3CA

In addition to identifying intragenic suppressors, we also discovered nine intergenic suppressors carrying missense mutations in the kinase domain of DYF-5 (Appendix Fig. S1A; Fig. EV5B). A reference null allele of *dyf-5(mn400)* exhibited comparable rescue effects on the ciliary phenotype of *osm-3CA* mutants (Appendix Fig. S1B,C). DYF-5 belongs to a conserved family of mitogen-activated protein kinases (MAPKs) that localize at the distal region of the cilium to regulate ciliary elongation (Berman et al, 2003; Burghoorn et al, 2007; Omori et al, 2010; Yi et al, 2018).

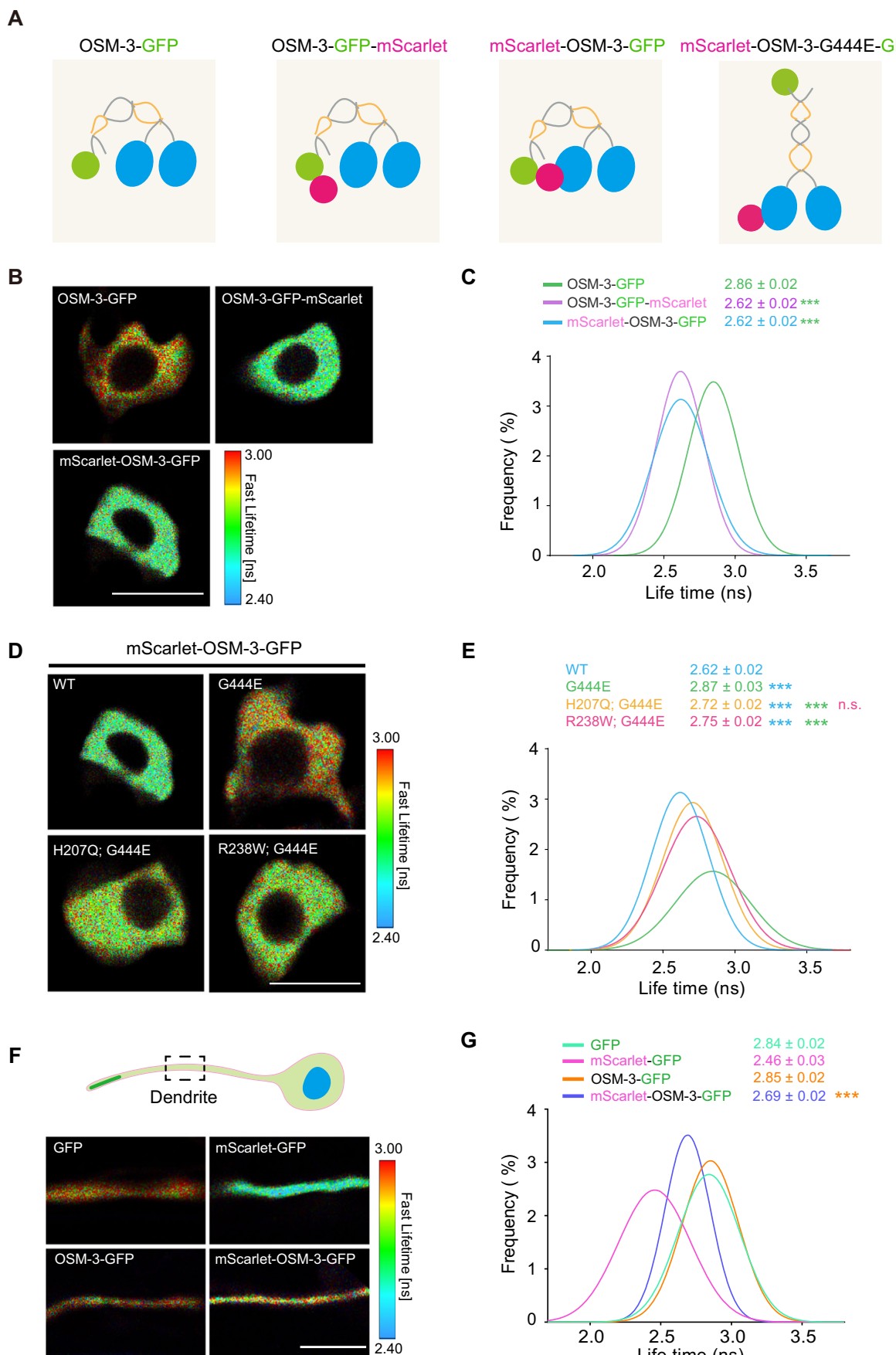

**Figure 6. Fluorescence lifetime imaging microscopy (FLIM) images of the WT and mutant OSM-3 proteins in sensory neuron soma.**

(A) Schematic diagram designed for FLIM imaging. (B) Representative FLIM images of WT OSM-3 with different fluorescence labeling. OSM-3::GFP, OSM-3::GFP::mScarlet and mScarlet::OSM-3::GFP are expressed under the control of P*dyf-1* in *osm-3(p802)* mutant animals. Fluorescence lifetimes of GFP are shown with a scale of pseudo color ranging from 2.40 to 3.00 nanoseconds (ns). Scale bar, 5 µm. (C) Frequency distribution of GFP fluorescence lifetimes measured from the images as shown in (B). Over $10^6$ events from 30 areas in 10 worms of each strain were measured. the plots were fit with a Gaussian distribution. Comparisons were performed between OSM-3::GFP and OSM-3::GFP::mScarlet or mScarlet::OSM-3::GFP. ***$P < 0.001$ by one-way ANOVA using BH method to adjust $P$ values. (D) Representative FLIM images of WT (image from (B)) or mutant OSM-3 whose N-terminus was tagged with mScarlet tag and C-terminus was tagged with GFP in sensory neurons. Fluorescent protein-tagged OSM-3 are expressed under the control of P*dyf-1* in *osm-3(p802)* mutant animals. Fluorescence lifetimes of GFP are shown with a scale of pseudo color ranging from 2.40 to 3.00 nanoseconds (ns). Scale bar, 5 µm. (E) Frequency distribution of GFP fluorescence lifetimes measured from the images as shown in (D). The statistical analysis was performed the same as in (C). Blue asterisks represent comparisons between WT OSM-3 and the other three mutant OSM-3. Green asterisks represent comparisons between OSM-3-G444E and OSM-3-H207Q-G444E or OSM-3-R238W-G444E. n.s. not significant, represents the comparison between OSM-3-H207Q-G444E and OSM-3-R238W-G444E. ***$P < 0.001$ by one-way ANOVA using BH method to adjust $P$ values. (F) Representative FLIM images in dendrites of sensory neurons in the strains as labeled. Scale bar, 5 µm. (G) Frequency distribution of GFP fluorescence lifetimes measured from (F). Over $5 \times 10^5$ events from 30 areas in 10 worms of each strain were measured. The statistical analysis was performed the same as in (C). Source data are available online for this figure.

Loss-of-function mutations in DYF-5 in *C. elegans* or its homologs in other species have been associated with excessively long cilia and retinitis pigmentosa (Berman et al, 2003; Burghoorn et al, 2007; Omori et al, 2010; Ozgul et al, 2011; Tucker et al, 2011). To investigate the relationship between DYF-5 and OSM-3CA, we expressed *dyf-5* cDNA in ciliated neurons of the *osm-3CA; dyf-5* double mutant. Remarkably, all examined transgenic animals displayed the same short ciliary phenotypes and reproduced the Dyf defects observed in *osm-3CA* animals ($n = 50$ transgenic animals) (Appendix Fig. S1B,C). This suggests that DYF-5 regulates OSM-3CA in a cell-autonomous manner, directly influencing its ciliary length and function.

Through TEM examinations, it became evident that *dyf-5; osm-3CA* double-mutant animals have regenerated the ciliary distal segment associated with singlet microtubules (Appendix Fig. S1H). Given that the CFAP-20 protein distinctly bridges the A and B singlet MTs to form doublets, it serves as a pivotal marker in discerning middle segment doublets from distal singlets (Chen et al, 2023; Chrystal et al, 2022; Yanagisawa et al, 2014). Leveraging the mScarlet-tagged CFAP-20 marker for additional analysis, we observed the red fluorescence's confinement to the middle segments, underscoring the notion that *dyf-5; osm-3CA* double mutants form distal segments (Appendix Fig. S1G–I).

Although the *osm-3CA* intragenic suppressors completely eliminated OSM-3CA::GFP ejecta and restored the ciliary localization of OSM-3CA::GFP (Fig. 5C; Appendix Fig. S5F), the reduction of OSM-3CA::GFP ejecta was not evident in *osm-3CA; dyf-5* double mutants. However, a small proportion of OSM-3CA::GFP was restored within the cilia, despite its immobility (Appendix Fig. S1B,S1F). Our in vitro protein kinase assays revealed that DYF-5 directly phosphorylates the C-terminal fragment of OSM-3 (444–699) (Appendix Fig. S1D,E), suggesting that DYF-5 kinase may act directly on OSM-3CA. In addition, DYF-5's ability to phosphorylate other IFT proteins (Jiang et al, 2022) indicates potential alternative mechanisms. It is conceivable that the absence of DYF-5-mediated phosphorylation could alter OSM-3 CA's conformation and activity, either directly or indirectly, thereby partially recovering its localization in the cilia.

## Conformational and activity regulation in the stepwise control of OSM-3CA fate

To further investigate the influence of protein conformation and activity on the fate of OSM-3CA, we utilized a kinesin rigor mutation.

This involved introducing the G234A mutation within the ATP-binding motifs of kinesin-1, thereby disrupting its ATPase activity and causing its tight binding to microtubules (Rice et al, 1999). To explore the impact of this mutation on OSM-3CA, we generated a recombinant full-length OSM-3::GFP protein carrying the analogous G235A substitution. We showed that OSM-3G235A was successfully bound to microtubules but did not demonstrate any gliding activity (Movie EV8). Subsequently, we introduced the G235A mutation into the genome of the OSM-3::GFP strain, and the resulting organisms displayed a loss of the distal ciliary segment, resembling the ciliary structure observed in *osm-3* null mutants (Fig. 7A). Importantly, the fluorescence signal of OSM-3G235A was confined to the remaining middle ciliary segment and did not form any granular structures in *C. elegans* (>100 animals examined) (Fig. 7A). Due to the rigor mutation, OSM-3G235A did not undergo IFT and likely exhibited strong binding to ciliary microtubules (Fig. 7A). To investigate the role of heterotrimeric kinesin-II in the ciliary localization of GFP-tagged OSM-3G235A, particularly in the remaining middle segment, we introduced a *klp-11* null allele, which impairs kinesin-II function, into the *osm-3(G235A)* strain. Our observations revealed that OSM-3G235A was absent from the cilia but accumulated at the ciliary base (Fig. 7A). This finding suggests that the observed presence of OSM-3G235A in the middle segment is not due to the inherent movement of OSM-3G235A itself but rather results from the import by kinesin-II.

Next, we introduced the G235A mutation into the OSM-3G444E::GFP strain. Interestingly, the GFP-tagged OSM-3G235AG444E double mutant did not exhibit detectable signals within the cilia or outside of neurons and did not form puncta (Fig. 7B,C). Instead, this OSM-3G235AG444E kinesin double mutant displayed specific localization along the axons of sensory neurons, which differed from the distribution patterns observed in the OSM-3G235A or OSM-3G444E single mutants (Fig. 7B,C). The presence of OSM-3G235A in a rigor state within the cilia suggests that its ciliary localization is independent of its motor activity (Fig. 7A). In other words, a ciliary kinesin lacking motor activity can still enter the cilia. The rigor mutation G235A potentially shields OSM-3CA from its predisposition to aggregation through robust microtubule binding. Conversely, our findings show that the H207Q mutation in OSM-3 fails to bind microtubules, as evidenced by the absence of microtubule binding in the microtubule gliding assay (Fig. 5G) and minimal microtubule-stimulated ATPase activity (Fig. 5F). Notably, worms with either the *osm-3(H207Q)* mutation or the *osm-3(H207Q;G444E)* combination, derived from

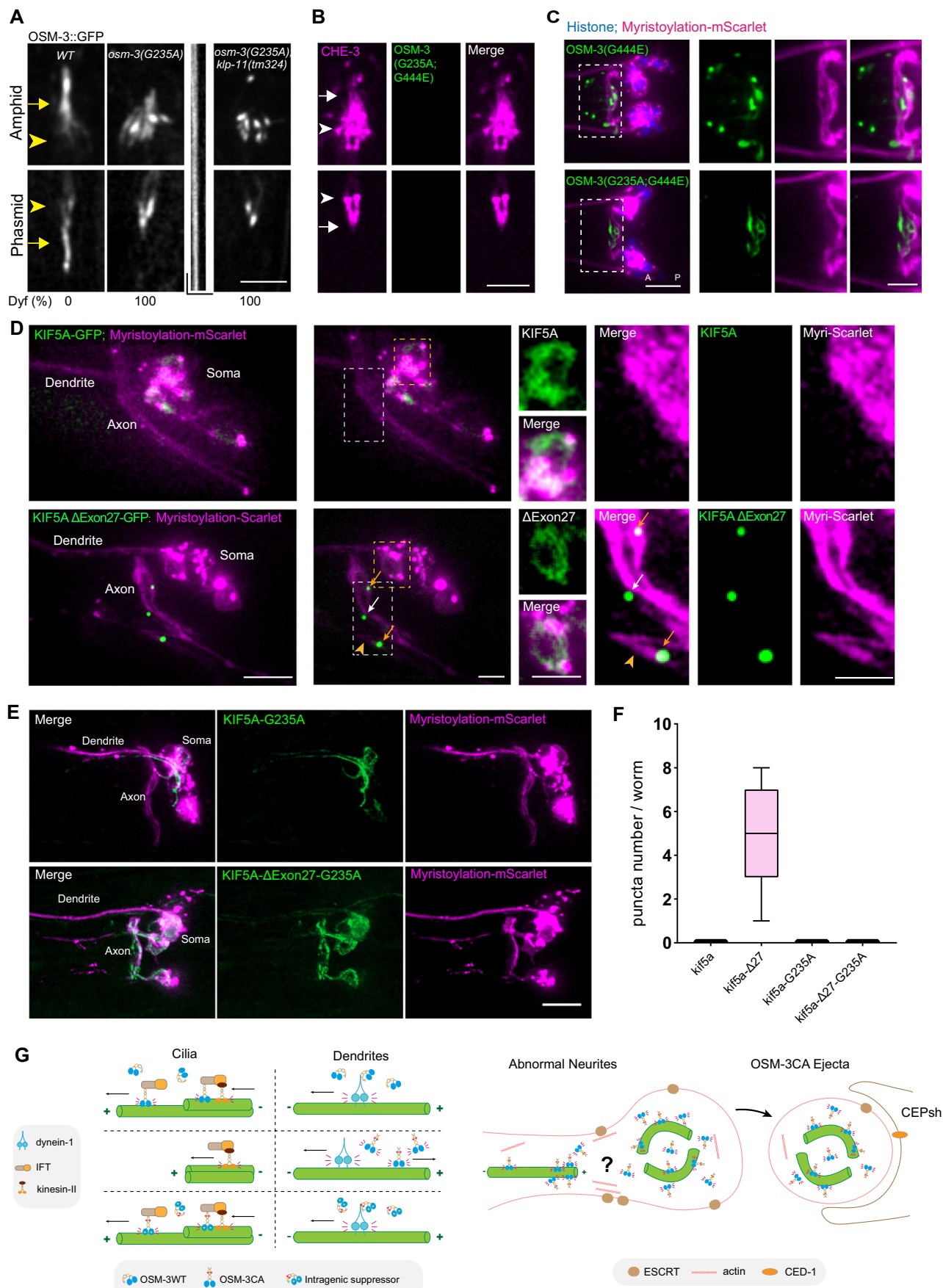

**Figure 7. Neurons dispose of hyperactive kinesin-1 resulting from an ALS-associated mutation.**

(A) Amphid and phasmid cilia in the WT, G235A mutant *osm-3(G235A)* (OSM-3-G235A::GFP KI animals) and *osm-3(G235A); klp-11(tm324)* double-mutant animals. Dyf, dye-filling defective; $N \geq 100$. Arrowheads indicate the ciliary base. Arrows indicate the junctions between the middle and distal segments. The WT panel is from Fig. 5C. Scale in image, 5 µm. Scales in kymograph, vertical, 5 s; horizontal, 5 µm. (B) The absence of ciliary localization of the endogenous GFP-tagged OSM-3-G235A-G444E in KI animals. The mScarlet-tagged endogenous CHE-3 is shown in magenta. Arrowheads indicate the ciliary base. Arrows indicate the junctions between the middle and distal segments. Scale bar, 5 µm. (C) Representative fluorescence images of the endogenous OSM-3-G444E or OSM-3-G235A-G444E localization relative to the ciliated sensory neurons. HIS-54::BFP and Myristoylation::mScarlet are expressed under the control of P*dyf-1* in OSM-3-G444E::GFP or OSM-3-G235A-G444E::GFP KI animals. Dashed boxes are enlarged on the right. The OSM-3-G444E panel is from Fig. 2C. Scale bars, 10 µm (left) and 5 µm (right). A anterior, P posterior. (D) Representative fluorescence images of the localization of KIF5A or KIF5AΔExon27 in sensory neurons. GFP-tagged WT KIF5A or the 27th exon deleted mutant KIF5A (KIF5AΔExon27) and the membrane marker Myristoylation::mScarlet are expressed under the control of P*dyf-1*. The dashed boxes (yellow: soma; white: axon) are enlarged on the right. Orange arrowheads, an ectopic neurite; orange arrows, the KIF5AΔExon27 granules colocalize with axon; white arrows, the KIF5AΔExon27 puncta that does not colocalize with axon. Scale bars, 5 µm. (E) Representative fluorescence images of the localization of KIF5A-G235A or KIF5A-ΔExon27-G235A in sensory neurons co-expressed with the membrane marker Myristoylation::mScarlet under the control of P*dyf-1*. Scale bar, 10 µm. (F) Statistics of puncta numbers of KIF5A mutants showed in (D, E). The center of the box is the mean value; the upper and lower bounds of the box are the first and third quartile, respectively; the whiskers show the minima and maxima. For each strain, $n > 15$ worms were analyzed. (G) A proposed model for the regulation of hyperactive OSM-3 in vivo. Source data are available online for this figure.

genetic screening or knock-in methods (Figs. 5C and EV5D,F), exhibited no punctate signals. Instead, their distribution was dispersed in the cilia and soma of ciliated neurons (Figs. 5C and EV5D,F). Given that the H207Q mutation results in a loss of ATPase activity without the heightened microtubule-binding capability observed in the G235A mutant, it implies that the ejection of OSM-3CA puncta is primarily driven by hyperactivity.

Furthermore, the absence of ciliary localization in the OSM-3G444E or OSM-3G235AG444E double mutants indicates that protein activity does not determine their entry into the cilia (Figs. 1F and 7C). Instead, it is the conformational changes induced by the G444E mutation that impede the ciliary localization of OSM-3. Conversely, the OSM-3G235AG444E double mutant remains confined to axons without being released, highlighting the requirement of hyperactivity for the disposal of OSM-3G444E (Fig. 7B,C). Therefore, neurons possess the ability to discern protein activity and selectively dispose of hyperactive kinesins, while cilia can recognize protein conformations and prevent the entry of kinesins with incorrect conformations, such as those that are extended and hyperactive.

### Neurons dispose of hyperactive kinesin-1 resulting from an ALS-associated mutation

A splicing mutation in the KIF5A gene of kinesin-1, linked to amyotrophic lateral sclerosis (ALS), leads to the exclusion of exon 27. This alteration gives rise to a mutant protein with a distinctive C-terminal sequence comprising 39 amino acid residues. (Brenner et al, 2018; Nicolas et al, 2018). Consequently, the mutant protein, confers a constitutively active state of KIF5A, exerts gain-of-function effects and induces neuronal toxicity (Baron et al, 2022; Nakano et al, 2022; Pant et al, 2022). Through a careful analysis of published data, we have discovered evidence suggesting that a fraction of the hyperactive KIF5A variants may be released into the culture medium (Appendix Fig. S2A) (Pant et al, 2022). Notably, we have also observed a similar release phenomenon in hyperactive KIF17, the mammalian counterpart of OSM-3, carrying an equivalent G444E mutation (Appendix Fig. S2B) (Hammond et al, 2010). Encouraged by these findings, we expressed GFP-tagged wild-type or hyperactive KIF5A in sensory neurons of *C. elegans*, and the membrane marker Myristoylation::mScarlet was expressed to label the periphery of neurons. While the GFP signal from wild-type KIF5A exhibited a homogeneous distribution within the cytoplasm of neurons

(Fig. 7D and Appendix Fig. S2C,D), the GFP signal from the hyperactive KIF5A displayed localization in the soma and also the puncta structures, some of which were observed outside the neurons (Fig. 7D,F; Appendix Fig. S2C,D). Furthermore, these neurons exhibited ectopic neurites posterior to the cell body (Fig. 7D, Appendix Fig. S2E). The release of puncta containing hyperactive KIF5A and the formation of aberrant neurites closely resemble the phenotypes associated with OSM-3CA. By expressing GFP-tagged hyperactive KIF5A under the control of mechanosensory neuron promoter P*mec-7*, we observed ejecta outside of the mechanosensory neuron as well (Appendix Fig. S2F,G), which suggests that the extracellular clearance of hyperactive kinesins may represent a widely employed mechanism by which living cells regulate kinesin hyperactivation.

We show that OSM-3CA carrying G444E mutation was expelled; however, OSM-3CA carries the rigor mutation G235A was not, raising the possibility that kinesin's ATPase activity might be pivotal for its expulsion. Supporting this hypothesis, we found that while wild-type KIF5A overexpression remained unexpelled, its ALS mutation variant was disposed (Fig. 7D,F; Appendix Fig. S2C,D). Moreover, introducing the rigor mutation into the ALS KIF5A ensured its retention within neuronal confines, thereby underlining the crucial role of motor activity in the disposal of hyperactive kinesin (Fig. 7E,F).

## Discussion

We propose a model that elucidates how neurons eliminate a hyperactive ciliary kinesin through clearance by glial cells (Fig. 7G). Following OSM-3CA production, sensory neurons detect its aberrantly extended conformation and prevent its localization and functional engagement within sensory cilia, where OSM-3 is typically situated and functions. The loss of ciliary localization converts this biochemically hyperactive kinesin into a state of functional loss in vivo, effectively mimicking the ciliary defects observed in the kinesin-null organism. Due to its continuous ATP hydrolysis, OSM-3CA likely generates abnormal forces and motility along microtubules, leading to cellular toxicity perhaps characterized by ATP overconsumption and interference with microtubule tracks. When such detrimental effects accumulate beyond the threshold that neurons can tolerate, neurons must eliminate hyperactive kinesins.

Future research will definitively ascertain whether GFP-tagged OSM-3CA puncta contain the hyperactive OSM-3 kinesin, pending the development of a reliable immunofluorescence protocol for *C. elegans* ciliated sensory neurons. Current time-lapse imaging of OSM-3CA::GFP puncta, as depicted in Figs. 2G and EV2C,D; Movies EV1–3, consistently exhibits rotational motion within these structures (Movie EV4). This finding, combined with the lack of similar rotational movement in puncta with GFP alone, strongly suggests the presence of active motor proteins within these GFP-tagged OSM-3CA puncta. Moreover, imaging of non-fluorescently-tagged OSM3-CA animals revealed no detectable green fluorescence in the nerve ring region, where OSM-3CA::GFP puncta typically localize (Fig. EV1C). These observations strongly indicate that the observed green fluorescence in the puncta originates from the fluorescently labeled mutant OSM-3CA protein, rather than GFP alone or nonspecific autofluorescence.

While existing publications have provided evidence for the analogous release of hyperactive kinesins KIF5-CA and KIF17-CA in mammalian systems (Hammond et al, 2010; Pant et al, 2022), the precise documentation and reasons for their extracellular localization remain elusive. Moreover, these motor proteins have predominantly been studied using GFP tags. Future investigations, therefore, should focus on examining endogenous hyperactive kinesins in mammalian neurons or patient samples, utilizing techniques like immunofluorescence or immunoblotting. Equally important is the development of a live imaging system in mammalian neurons to reveal the mechanisms underlying the release of the pathogenic KIF5A-CA kinesin. Such studies are imperative for clinically interpreting our findings and understanding the broader implications of hyperactive kinesin behavior.

Previous studies revealed that most of microtubules in the dendritic distal region are likely oriented towards the cell body, facilitating transport by the minus-end-directed dynein-1 motor protein of ciliary components from the soma through dendrites to the ciliary base (Li et al, 2017). Consequently, it is improbable that the plus-end-directed motor protein OSM-3CA kinesin would actively traverse along the dendrite. Moreover, the recombinant OSM-3CA protein displayed no inclination towards in vitro condensation or aggregation, facilitating our analysis of its motility characteristics in vitro. We hypothesize that OSM-3CA is more likely to travel into the axon, progressing toward the axon tip where microtubules predominantly exhibit plus-end orientation (Li et al, 2017). Accumulation of excessive OSM-3CA at the axon tip might lead to its local release, potentially as a cellular response to the excess of this hyperactive motor protein.

Based on our findings, the internalization and clearance of OSM-3CA by glial cells involve the machinery associated with apoptotic cell degradation. We hypothesize that the disposed OSM-3CA ejecta may expose phosphatidylserine on their inner membrane leaflet, which serves as an "eat-me" signal (Nagata, 2018) recognized by engulfment receptors on the plasma membrane of glial cells. Subsequently, glial cells internalize the OSM-3CA ejecta and utilize their degradation system to clear them. While our evidence supports the involvement of CED-1 in OSM-3CA clearance, we acknowledge the possibility that OSM-3CA might not follow the exact same pathway as apoptotic cell clearance. Notably, exopher degradation requires CED-1 but does not involve phosphatidylserine externalization (Melentijevic et al, 2017; Wang et al, 2023).

Similar to the repurposed engulfment pathway, we propose that hyperactive kinesin assemblies may utilize the neuronal actin cytoskeleton and membrane abscission system for their release. We observed the enrichment of OSM-3CA at the tips of aberrant neurites, which are not typically present in wild-type animals. These neurites appear to branch off from existing axons. The mechanism by which hyperactive kinesin leads to the formation of these additional neurites is mysterious, as the extension of the growth cone through Arp2/3-dependent actin nucleation events at the leading edge is considered a prerequisite for axon or dendrite formation (Dent et al, 2011; Xie et al, 2021; Zhu et al, 2016). We speculate that neurons may detect the abnormal forces generated by OSM-3CA, which in turn triggers actin polymerization. The mechanism of how the ejecta be produced and released out of neurons remains unclear. Our data indicate actin and the ESCRT complex may be involved for membrane abscission during OSM-3CA ejecta production.

While our investigations were largely rooted in *C. elegans* neurons, our rationale for expressing ALS kinesin in this model stemmed from the conspicuous extracellular localization of not just ALS kinesin but also hyperactive KIF17 kinesin as puncta (Hammond et al, 2010). Considering the different microtubule anatomy of human neurons and the *C. elegans* chemosensory neurons, it remains to an open question whether human neurons also employ the ejecta mechanism to expel hyperactive KIF5A associated with ALS. Nevertheless, future endeavors are geared towards corroborating these observations, determining if the exclusion of hyperactive kinesin overexpression is a prevalent phenomenon both in cultured cells and under pathological settings.

Taken together, our study introduces valuable ideas: A biochemical hyperactive kinesin behaves as a complete loss-of-function protein in vivo because neurons eliminate it by transferring it to glial cells for clearance. Notably, the canonical kinesin KIF5A, instrumental in the translocation of a myriad of neuronal cargo molecules, manifests a myriad of neuronal disorders, including ALS, upon its functional abrogation (Brenner et al, 2018; Naruse et al, 2021; Nicolas et al, 2018). The ALS-associated kinesin mutation, localized to the C-terminal cargo-binding domain, was deemed a loss-of-function variant (Brenner et al, 2018; Naruse et al, 2021; Nicolas et al, 2018). Yet, rigorous biochemical and biophysical assays unveiled the ALS mutant kinesin as hyperactive rather than inactive (Baron et al, 2022; Nakano et al, 2022; Pant et al, 2022). Our results suggest that the expulsion of the hyperactive kinesin as an ejecta from neurons, inducing a consequential functional deficiency, suggesting that the pathology may result, at least in part, from its loss of function in vivo. This, in turn, may require a rethinking of therapeutic strategies for the treatment of diseases associated with hyperactive proteins.

# Methods

## *C. elegans* strain culture

*C. elegans* strains were cultured on nematode growth medium (NGM) plates at 20 °C. *Escherichia coli* strain OP50 was seeded on the plates as food of the worms. Appendix Tables S1–S3 summarize the strains, primers and plasmids used in this study.

## Molecular biology and genetics

For the generation of *osm-3* point mutations, target of CRISPR-Cas9: GACGATCAACGAGGATCCAA (AGG) was inserted into the pDD162 vector (Addgene #47549) by PCR linearizing. After DpnI digestion, the resulting PCR products were transformed into *E. coli* to generate plasmid. Homology recombination (HR) templates were constructed by cloning the 2069 base pairs (bp) homology arm that contains the rigor point mutation into pPD95.77 plasmids by using In-Fusion Advantage PCR cloning kit (Cat. # 639621, Palo Alto, CA, USA, Clontech). Primers used are listed in Appendix Table S1. The target sequence was selected by the CRISPR design tool (http://crispr.mit.edu). CRISPR/Cas9 constructs and HR templates together with *rol-6*[su1006] and P*odr-1*::dsRed markers were microinjected into the germ line of young adult OSM-3::GFP knock-in hermaphrodites at a concentration of ~50 ng/μL. The mutant strains were confirmed by PCR and Sanger sequencing.

For the transgenic strains, the overexpression constructs were generated by using PCR fusion-based approach. For OSM-3 and OSM-3 mutants, The genomic sequences of OSM-3, OSM-3(G444E), OSM-3(H207Q-G444E), OSM-3(R238W-G444E) were inserted into the pDONR vectors that contain the *dyf-1* promoter and GFP/mScarlet::*unc-54* 3' UTR. For HIS-54 and Myristoylation, their coding sequences were inserted into the pDONR vectors that contain the *dyf-1* promoter and BFP/mScarlet::*unc-54* 3' UTR. The *hlh-17* promoter was inserted into the pDONR vector that contains mScarlet::*unc-54* 3' UTR to overexpress mScarlet in CEPsh glia. For CED-1, The genomic sequences of CED-1 were inserted into the pDONR vectors that contain *ced-1*, *hlh-17* or *Y37A1.B* promoter and *unc-54* 3' UTR. For DYF-5, the coding sequences were inserted into the pDONR vectors that contain the *dyf-1* promoter and BFP/mScarlet::*unc-54* 3' UTR. Primers used are listed in Appendix Table S2. For KIF5A, the WT or Δ27th exon coding sequences of mouse kif5a (the relevant sequence of human 27th exon was deleted then the novel C-terminal sequence was added) were inserted into the pDONR vectors that contain the *dyf-1* promoter and *unc-54* 3' UTR. The overexpression constructs together with *rol-6*[su1006] were microinjected into the germ line of young adult hermaphrodites at a concentration of 15–20 ng/μL. Marker-positive F1s were singled and cultured. The F2s inherited the transgenes were identified as the transgenic lines. At least two independent transgenic lines were maintained and used for experiments in this study.

To isolate suppressors of *osm-3(G444E)*, *osm-3(G444E)*-GFP knock-in worms were synchronized to L4 stage and treated with ethyl methanesulfonate (EMS). Gravid P0 worms were bleached and F1 worms were cultured under standard conditions. Adult F2 animals were subjected to Dye-Filling Assay and Dye-positive mutant animals were individually cultured. Whole-genome sequencing was performed to all suppressor strains to identify candidate genes.

## Live-cell imaging

The procedures were performed as described before (Chai et al, 2012). Worms at different developmental stages were anesthetized with 0.1 mmol/L levamisole in M9 buffer and fixed on 3% agarose pads at 20 °C. The equipment and used for live-cell imaging were the same as before (Xie et al, 2020). The microscope was equipped with a 100×, 1.49 numerical aperture (NA) objective. 405-, 488-, and 561-nm lines of a Sapphire CW CDRH USB Laser System was attached to a spinning disk confocal scan head (Yokogawa CSU-X1 Spinning Disk Unit). Images were acquired by μManager (https://www.micro-manager.org) at an exposure time of 200 ms. And we processed and quantified the images with ImageJ software. We used FV1200 (Olympus) equipped with PicoHarp 300 (PicoQuant) to perform fluorescence lifetime imaging. Data was analyzed by SymPhoTime 64 software.

## Dye-filling assay

As previously described (Xie et al, 2020), young-adult worms of different genotypes were collected with 1 ml M9 solution in 1.5-ml tube separately. Then, dyes (DiI, 1,1'-dioctadecyl-3,3,3',3',-tetra-methylindocarbocyanine perchlorate; Sigma-Aldrich, St. Louis, MO, USA) were added at a 20 μg/ml final concentration. After incubation in the dark for 30 min at room temperature, worms were transferred to OP50 seeded NGM plates. Dye-filling ratios of the worms were examined 2 h later via fluorescence stereoscope.

## FRAP

FRAP experiments are performed using an Airyscan confocal microscope (LSM900, Carl Zeiss) equipped with a 63×1.4 NA objective. 488-nm laser at 100% power is used to photobleach selected regions, and images are acquired with 1% power. Fluorescence recovery curves in Fig. 4 are fitted with an exponential equation: $I(t) = I_0 + (I_R-I_0)(1-e^{-kt})$, which $I(t)$ represents the fluorescence intensity at time $t$ after photobleaching, $k$ is the constant describing the rate of recovery, $I_0$ is the fluorescence intensity immediately after the photobleaching, and $I_R$ is the maximum recovered fluorescence. The recovery half-time is calculated by $t_{1/2} = ln2/k$. Similarly, the fluorescence decay curve in Fig. 4E is fitted with $I(t) = I_P + (I_0-I_P)·e^{-Kt}$, which $I(t)$ represents the fluorescence intensity at time $t$, $K$ is the constant describing the rate of fluorescence decay, $I_R$ is the fluorescence intensity after it reaches the plateau, $I_0$ is the fluorescence intensity right before photobleaching. The half-time of fluorescence decay is calculated by $T_{1/2} = ln2/K$.

## Transmission electron microscopy

We adopted the protocol established before (Xie et al, 2020). Young-adult animals on NGM plates were collected with M9 solution. After centrifugation, worms were loaded into a 50-μm deep specimen carrier. Via the Leica EM HPM100 high-pressure freezing system, animals in the carriers were frozen. Then we transferred the carriers into the fixative (acetone solution of 1% osmium tetroxide and 0.1% uranyl acetate) in a microcentrifuge tube under liquid nitrogen condition. Worms were fixed following the standard fixation program in Leica EM AFS2 machine. Then we washed the fixed specimens with pure acetone for three times. Using SPI-PON 812 resin, specimens were infiltrated. After polymerization at 60 °C, specimens were made to 90-nm ultrathin sections via Leica EM UC7 Ultramicrotome. We picked the

sections on 200 mesh copper grids and post-stained them with Reynold's lead citrate and 2% uranyl acetate. TEM Images were finally obtained via FEI Tecnai G2 Spirit (120 kV) electron microscope.

## Protein preparation

cDNA of EGFP tagged WT OSM-3 was cloned into pET.M.3 C plasmid for prokaryotic expression. Point mutations were generated on the WT OSM-3 construct. The expression and purification of all OSM-3 followed the published methods (Case et al, 1997; Imanishi et al, 2006). Protein concentration was determined by BCA assay.

## In vitro single molecule, microtubule gliding, and MT-stimulated ATPase assays

The assays were performed as described previously (Case et al, 1997; Imanishi et al, 2006; Mohamed et al, 2018). In brief, to track the single-molecule movement of eGFP-fused motors, microtubules were attached on a flow cell by antibodies (SAP4G5), and appropriate concentration of purified motors were flowed in with assay buffer (BRB80, 1 mM ATP/Mg$^{2+}$, 1% β-Mercaptoethanol, 0.08 mg/mL glucose oxidase, 0.032 mg/mL catalase and 80 mM glucose); eGFP signals were visualized by Olympus IX83 microscopy equipped with a 150× (numerical aperture (NA) 1.45 oil, Olympus) objective lens and an ORCA-Flash4.0 V3 camera. The system was controlled by Micro-Manager 2.0. For MT gliding assays, motors were attached to the surface of a flow cell via pre-coated GFP antibodies and gliding of rhodamine-labeled microtubules were recorded by TIRF microscopy. Microtubule stimulated ATPase activity assays were performed with a commercial kit (HTS Kinesin ATPase Endpoint Assay Biochem Kit, Cytoskeleton Inc.) following the manufacturer's instructions.

## RNA sequencing and editing analysis

Using our previously developed protocols (Li et al, 2021), we cultured the synchronized worms on NGM plates without OP50 bacteria. At L1 larval stage, worms were harvested and lysed with TRIzol reagent (Invitrogen). According to the manufacturer's protocol, we extracted total RNA of the worms. We quantified RNA concentration with the Qubit RNA High Sensitivity Assay Kit (Invitrogen) and assessed RNA quality with Agilent 2100 bioanalyzer system. Using the KAPA RNA HyperPrep Kit, 50–500 ng total RNA was prepared for the construction of RNA library (RNA integrity number >6.0). Library samples were analyzed and quantified with Agilent 2100 bioanalyzer system and Qubit. Using the Illumina HiSeq platform, samples were finally sequenced, generating the 150-bp paired-end reads. Using the RNA-seq reads, we performed RNA editing analysis according to the method we published before (Li et al, 2021).

## AlphaFold prediction

OSM-3 structure was predicted by AlphaFold Multimer using ColabFold (Evans et al, 2022; Jumper et al, 2021; Mirdita et al, 2022), with amber refinement and the use of templates.

## Quantifications and statistical analysis

We used ImageJ software to perform measurements and quantifications. For cilium length, the indicated numbers of amphid and phasmid cilia were randomly measured. For the quantifications of OSM-3-G444E granules, we used "Analyze Particle" tool of ImageJ to identify and mark all the granules in a worm with a consistent threshold. Then we performed measurement, granule number and size were outputted together. The fluorescence intensity of individual granule was measured without discrimination. For IFT velocity, the indicated numbers of IFT particles in amphid and phasmid cilia were randomly selected for the measurement. For FLIM acquisition, ten transgenic worms for each strain were recorded for analysis. Briefly, GFP lifetime in the soma of sensory neurons were acquired, and three regions in each soma were randomly selected as ROI for calculation for one worm. Microtubule-stimulated ATPase activities were derived from three assays and the average activity of WT OSM-3 was set to 100%. Single molecule and gliding data were measured using ImageJ software and all the events measured were selected randomly. After measurements and quantifications, we performed statistical analysis in GraphPad Prism. The statistical differences were determined by two-tailed Student's t-test analysis as indicated in the figure legends. The frequency distribution of IFT velocity and GFP lifetime was analyzed and fitted with a Gaussian distribution curve.

## Data availability

This study includes no data deposited in external repositories.

The source data of this paper are collected in the following database record: biostudies:S-SCDT-10_1038-S44318-024-00118-0.

## Peer review information

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

## Acknowledgements

This work was supported by the National Natural Science Foundation of China (grants 31991191, 32200612, 32071191, and 31971160); the National Key R&D Program of China (2019YFA0508401); Strategic Priority Research Program of CAS (XDB37020302); Deutsche Forschungsgemeinschaft (DFG) grant OE 501/5-1 (Project ID 449713185). C Xie was supported by the CLS Postdoctoral Fellowship.

## Author contributions

**Chao Xie**: Conceptualization; Data curation; Formal analysis; Funding acquisition; Investigation; Writing—original draft; Writing—review and editing. **Guanghan Chen**: Conceptualization; Data curation; Formal analysis; Investigation; Writing—original draft; Writing—review and editing. **Ming Li**: Data curation; Formal analysis. **Peng Huang**: Data curation. **Zhe Chen**: Data curation. **Kexin Lei**: Data curation. **Dong Li**: Data curation. **Yuhe Wang**: Data curation. **Augustine Cleetus**: Data curation. **Mohamed AA Mohamed**: Data curation. **Punam Sonar**: Data curation. **Wei Feng**: Supervision; Methodology. **Zeynep Ökten**: Supervision; Methodology. **Guangshuo Ou**: Conceptualization; Supervision; Funding acquisition; Methodology; Writing—original draft; Writing—review and editing.

Source data underlying figure panels in this paper may have individual authorship assigned. Where available, figure panel/source data authorship is listed in the following database record: biostudies:S-SCDT-10_1038-S44318-024-00118-0.

## Disclosure and competing interests statement

The authors declare no competing interests.

# Expanded View Figures

**Figure EV1. The OSM-3-G444E mutation is recessive and loss-of-function, related to Fig. 1.**

(A, B) Frequency distribution of single-molecule velocity of WT and G444E mutant OSM-3. The velocity curve was fitted with a Gaussian distribution. *n* number of single-molecule events measured, v velocity. Data are mean ± SD. (C) Representative image of non-fluorescently-tagged OSM-3-G444E in *osm-3(sa125)*, yellow dashed lines showed the pharynx, A, anterior; P, posterior. Scale bar, 10 µm. (D) Heterozygotic GFP-KI animals expressing one copy of WT *osm-3* and the other copy of *osm-3(G444E)*. Cilia that are visualized with mScarlet-tagged endogenous DYF-11 are indistinguishable from WT animals. Arrowheads indicate the ciliary base. Arrows indicate the junctions between the middle and distal segments. Scale bar, 5 µm. (D) shows GFP fluorescence in cilia whereas (E) shows GFP puncta around axons or ejecta outside of sensory neurons. Scale bar, 10 µm. (F) Expression of the WT *osm-3* gene under the control of a ciliated neuron-specific promoter P*dyf-1* did not rescue the ejecta formation in *osm-3 (G444E)* mutants. Scale bar, 10 µm. (G) Expression profiles at *osm-3* locus in WT and *osm-3(G444E)* strains. Gene body schematic is shown at the bottom. (H) Representative RNA editing analysis at *osm-3* locus in WT and *osm-3CA* strains. No editing events were found. The bar showing editing level and gene body schematic are shown above.

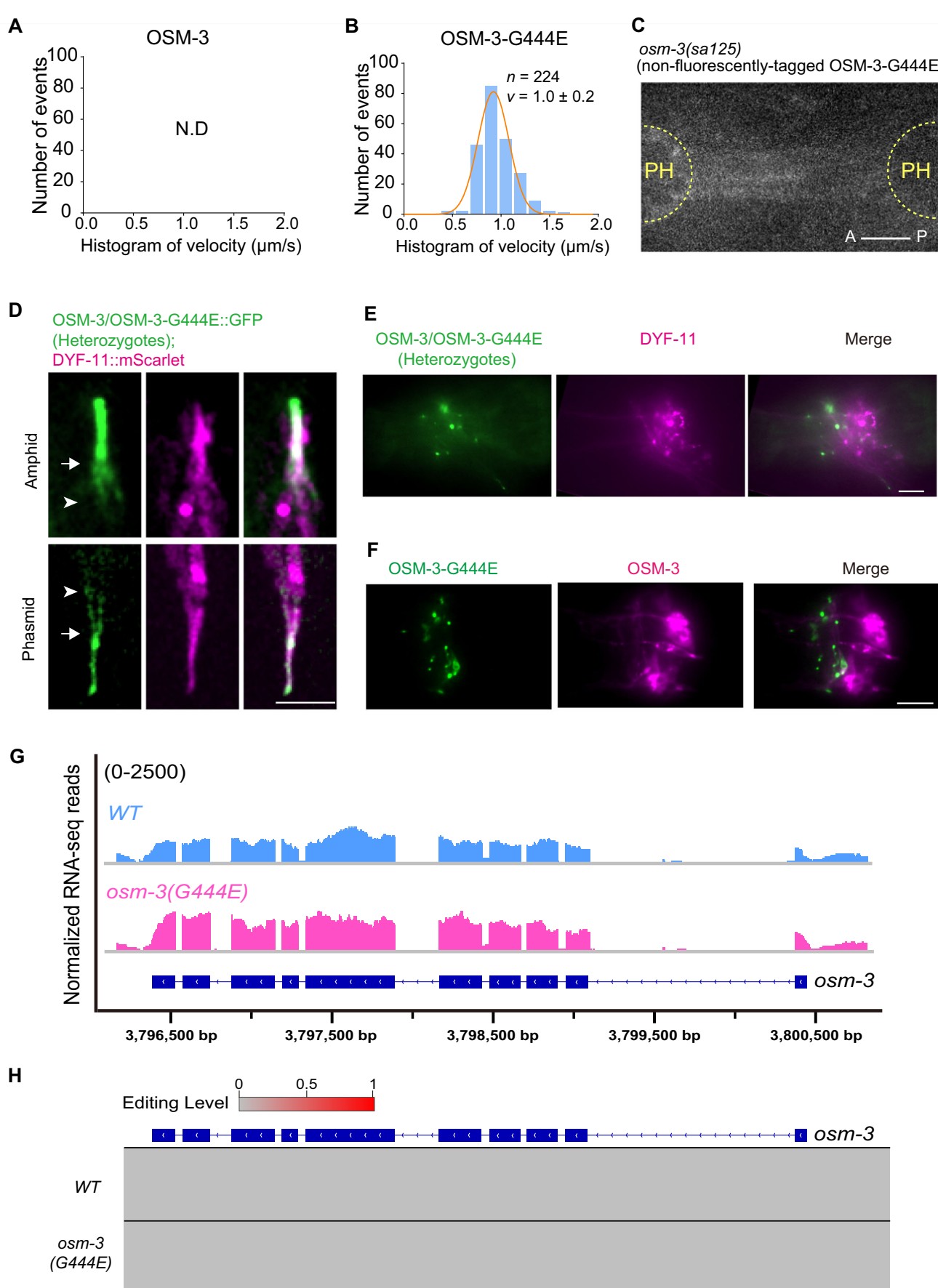

**A**

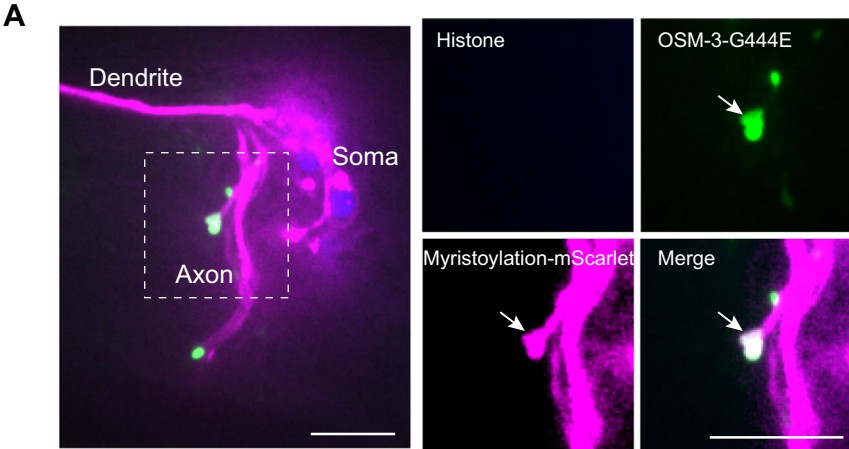

**B**

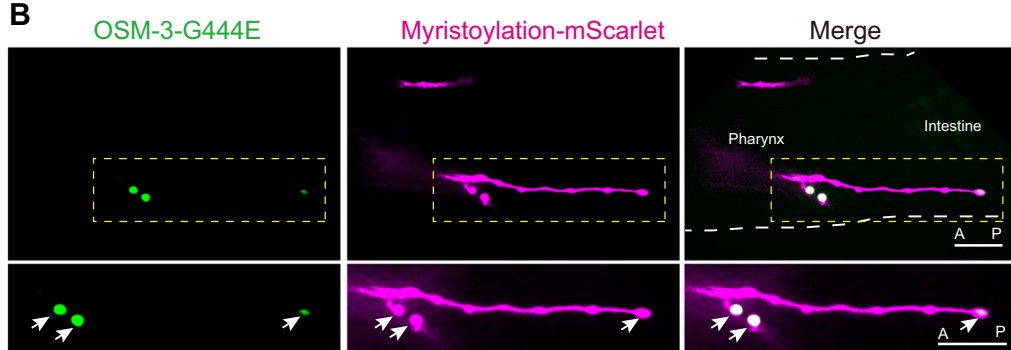

**C**

OSM-3-G444E; Myristoylation-mScarlet

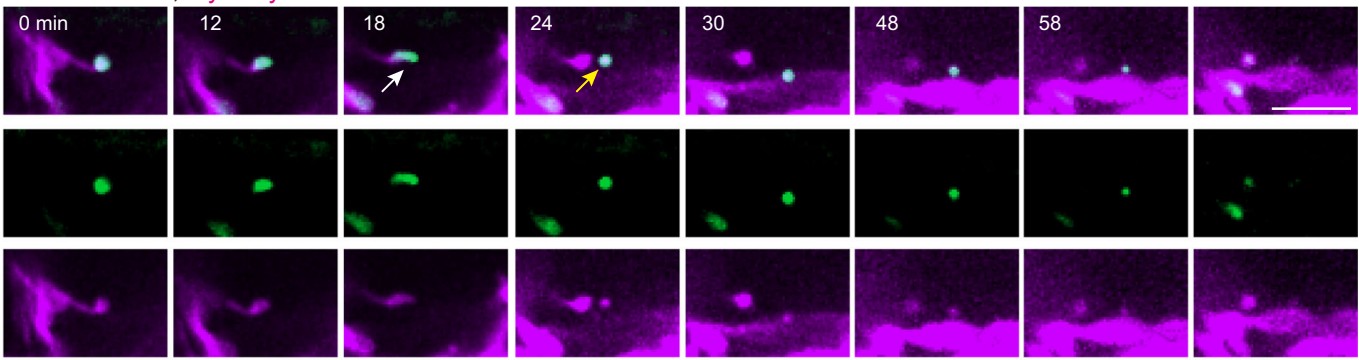

**D**

OSM-3-G444E; Myristoylation-mScarlet

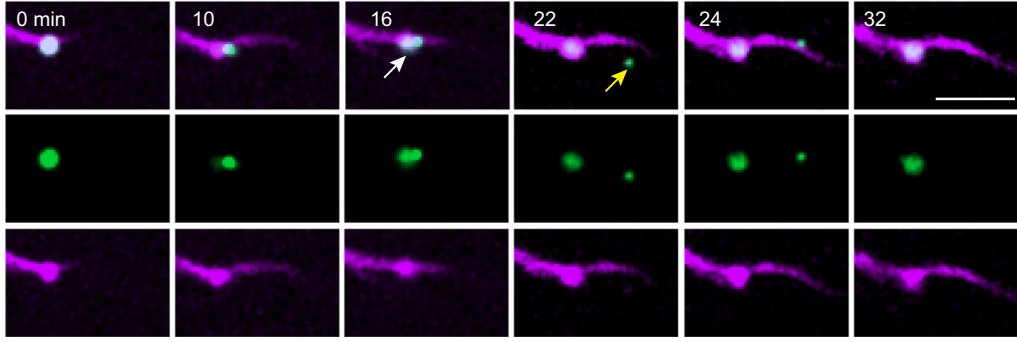

**Figure EV2.   Localization and disposal of hyperactive OSM-3 at the tips of ectopic neurites, related to Figs. 2 and 3.**

(A, B) Representative fluorescence images of OSM-3-G444E granules at the tips of axons or ectopic neurites. HIS-54::BFP and Myristoylation::mScarlet are expressed under the control of P*dyf-1* in OSM-3-G444E::GFP KI animals. Dashed box is enlarged on the right (A) or the bottom (B). Arrows, the tip of the neurites where the hyperactive OSM-3 localized. Scale bars, 10 μm. A anterior, P posterior. (C, D) Fluorescence time-lapse images of endogenous OSM-3-G444E in the sensory neurons. White arrows, the punctum that is about to be released; yellow arrows, the released punctum. Scale bars, 5 μm.

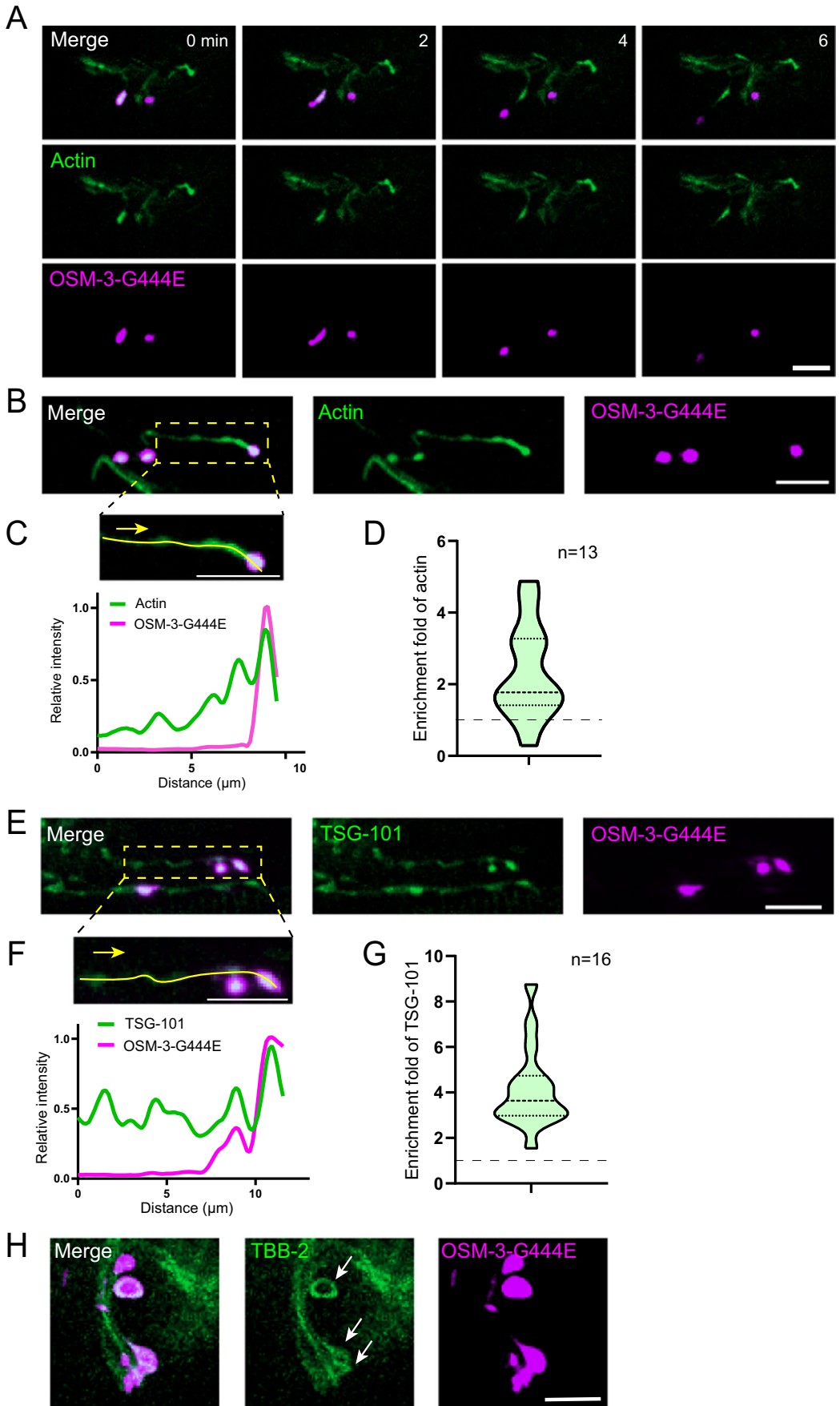

**Figure EV3. Colocalization of OSM-3-G444E ejecta with actin, ESCRT I component TSG-101 and microtubules.**

(A) Fluorescence time-lapse images of OSM-3-G444E and its colocalization with actin in the sensory neurons. GFP-tagged MoesinABD is expressed under the control of Pdyf-1 in OSM-3-G444E-mScarlet KI worms. Scale bar, 5 μm. (B) Representative fluorescence images of OSM-3-G444E puncta that colocalize with actin. Dashed box is enlarged in (C). Scale bar, 5 μm. (C) Line scan shows colocalization of OSM-3-G444E-mScarlet and MoesinABD labeled actin. Relative intensities are plotted using normalized gray values along the scanning line. Yellow arrow shows the scanning direction on the scanning line. Scale bar, 5 μm. (D) Enrichment fold of MoesinABD labeled actin. Enrichment fold = [(Mean Gray Value of Colocalized Area)-(Background Gray Value)]/ [(Mean Gray Value of Non-Colocalized Neurites)-(Background Gray Value)]. *n* number of events analyzed. (E) Representative fluorescence images of OSM-3-G444E puncta that colocalize with GFP-tagged TSG-101. Dashed box is enlarged in (F). Scale bar, 5 μm. (F) Line scan shows colocalization of OSM-3-G444E-mScarlet and GFP::TSG-101. Relative intensities are plotted using normalized gray values along the scanning line. Yellow arrow shows the scanning direction on the scanning line. Scale bar, 5 μm. (G) Enrichment fold of GFP::TSG-101. *n* number of events analyzed. (H) Representative fluorescence images of OSM-3-G444E puncta that colocalize with TBB-2, TBB-2 forms ring-like structures at the edges of the puncta. Scale bar, 5 μm.

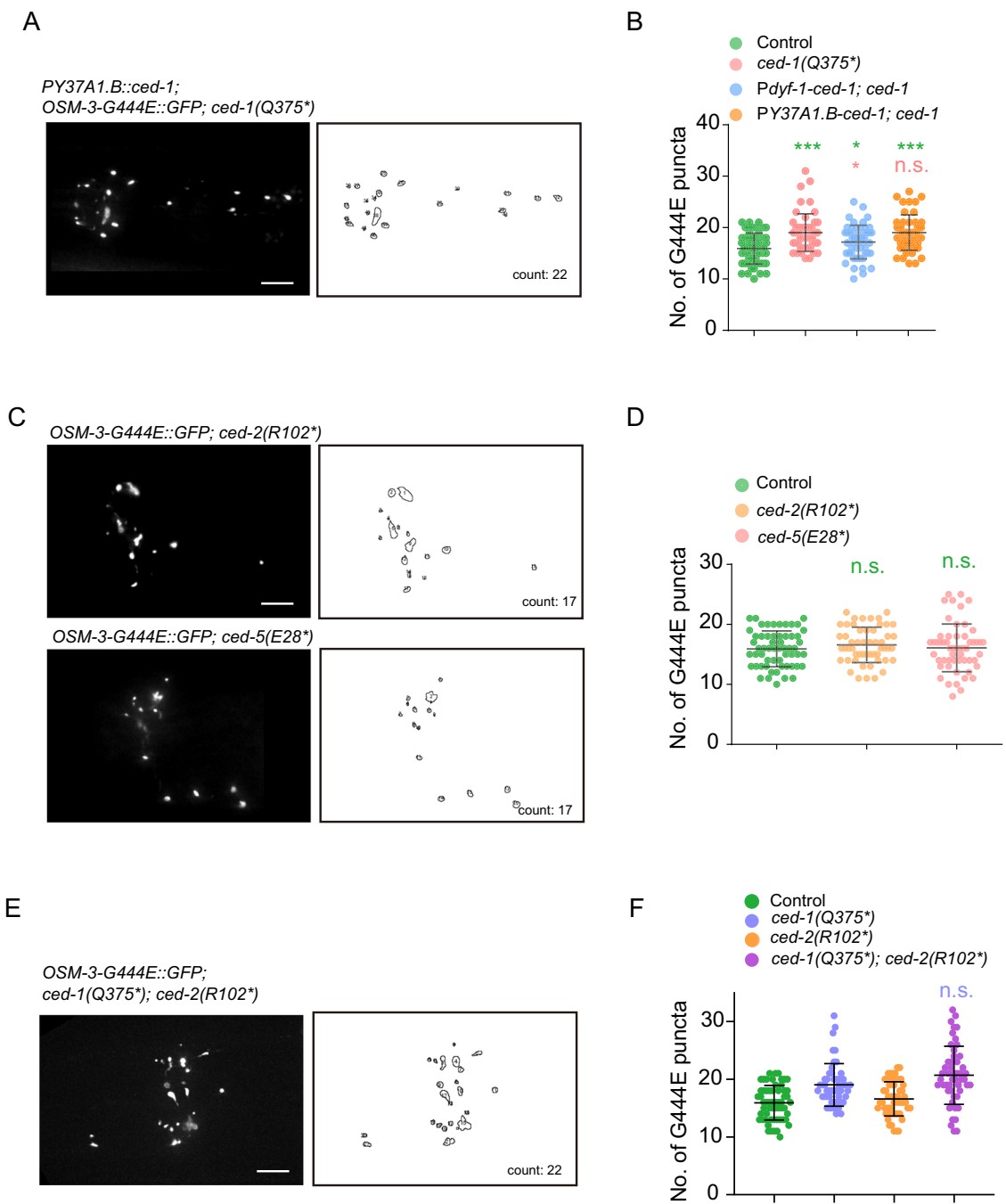

**Figure EV4.  *ced-2* and *ced-5* are not involved in the clearance of OSM-3-G444E ejecta.**

(A) Representative fluorescence image of OSM-3-G444E ejecta in *ced-1(Q375Stop)* mutants with the overexpression of wild-type *ced-1* under the control of PY37A1.B. Count, the number of ejecta. Scale bar, 10 μm. (B) Quantification of the number of OSM-3-G444E ejecta in the strains shown in (A), *n* > 50 worms were analyzed for each strain. Data are mean ± SD. (C) Representative fluorescence image of OSM-3-G444E ejecta in *ced-2(R102Stop)* or *ced-5(E28Stop)* strains. (D) Quantification of the number of OSM-3-G444E ejecta in the strains shown in (C), *n* > 50 worms were analyzed for each strain. Data are mean ± SD. (E) Representative fluorescence image of OSM-3-G444E ejecta in *ced-1(Q375Stop); ced-2(R102Stop)* double mutant. (F) Quantification of the number of OSM-3-G444E ejecta in the strains shown in (E), *n* > 50 worms were analyzed for each strain. Data are mean ± SD. ***$P < 0.001$, *$P < 0.05$, n.s. not significant, by one-way ANOVA using BH method to adjust *P* values.

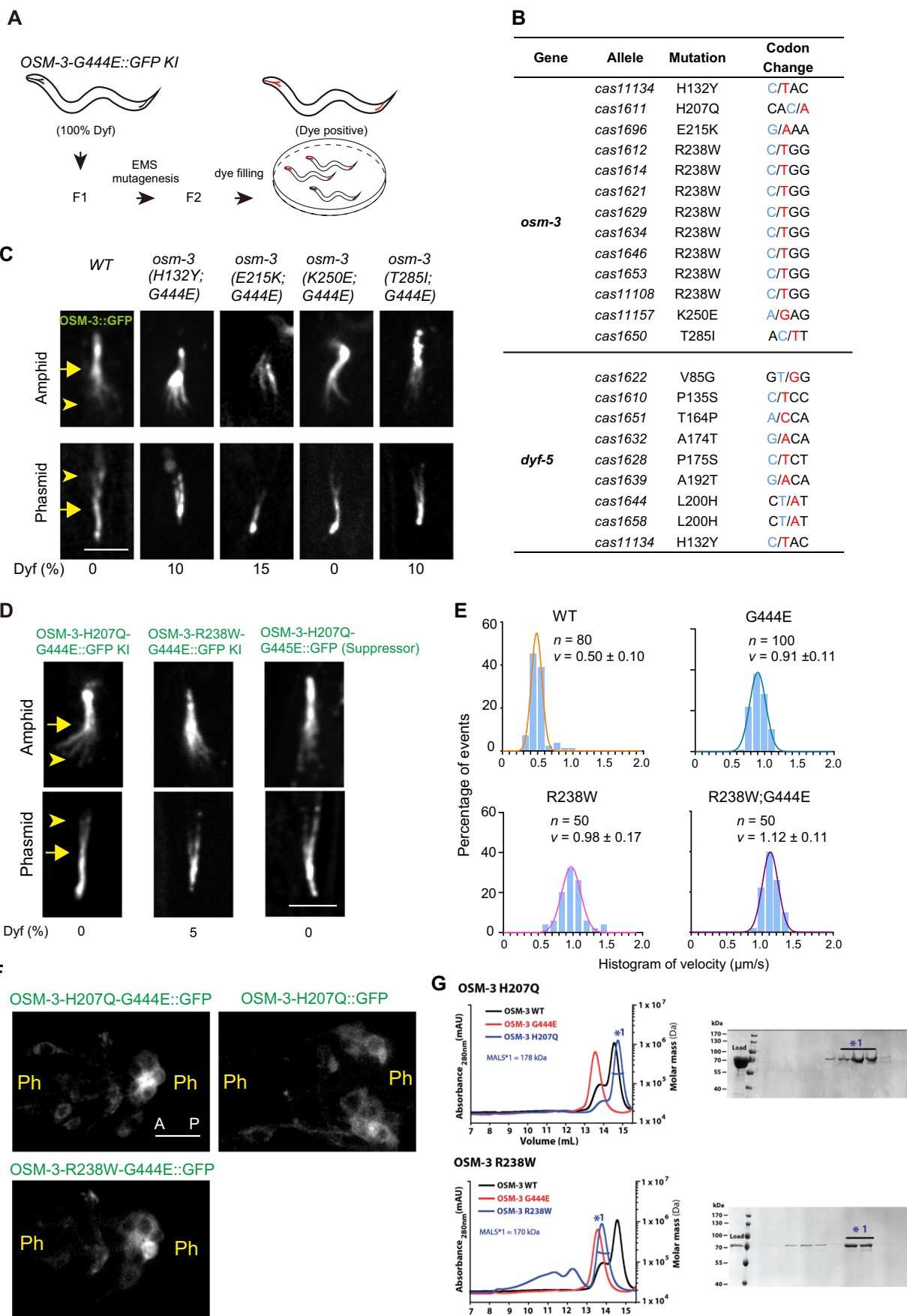

◀ **Figure EV5.  Intragenic mutations that rescue the ciliary phenotypes in *osm-3(G444E)* mutant and restore the localization of endogenous OSM-3-G444E.**

(A) Flowchart of the suppressor screening. OSM-3-G444E::GFP KI animals at the late L4 stage were treated with ethyl methanesulfonate (EMS). By filling with the fluorescent dye DiI, dye-positive F2 progenies were considered putative suppressors. About $10^5$ haploid genomes were screened in 5 rounds of screenings. The localization of OSM-3-G444E were examined via confocal microscopy. Mutant genes were cloned by the whole-genome sequencing. (B) Molecular lesions of the *osm-3* and *dyf-5* suppressor alleles. (C) Amphid and phasmid cilia in WT and the suppressors harbor the mutations shown in Fig. 5A, B. GFP-tagged endogenous WT OSM-3 or the corresponding mutant OSM-3 are shown. Dyf, dye-filling defective; N ≥ 100. Arrowheads indicate the ciliary base. Arrows indicate the junctions between the middle and distal segments. The WT image is from Fig. 5C. Scale bar, 5 μm. (D) Distribution of endogenous OSM-3-H207Q-G444E (left), OSM-3-R238W-G444E (middle) and OSM-3-H207Q-G445E (right) in sensory cilia. For the strain shown on the left and in the middle, H207Q or R238W mutation was generated in OSM-3-G444E::GFP KI animals by CRISPR/Cas9-triggered genome editing. The OSM-3-H207Q-G445E strain shown on the right was obtained from a suppressor screen restoring ciliary defects caused by H207Q mutation in OSM-3-H207Q::GFP KI worms. Dyf, dye-filling defective; N ≥ 100. Arrowheads indicate the ciliary base. Arrows indicate the junctions between the middle and distal segments. Scale bar, 5 μm. (E) Frequency distribution of MT gliding velocities showed in Fig. 5G. The velocity distribution curves were fitted with a Gaussian distribution. *n* number of MTs measured, v velocity. Data are mean ± SD. (F) Distribution of the endogenous OSM-3-H207Q-G444E (top left), OSM-3-H207Q (top right) and OSM-3-R238W-G444E (bottom left) in the soma of sensory neurons. Ph, pharynx. Scale bar, 10 μm. A anterior, P posterior. (G) Left, overlays of the elution profiles of WT (line in black), G444E (red) and H207Q or R238W mutant (blue) OSM-3. The molar mass of OSM-3-H207Q or R238W determined from the MALS fit is shown on the left; right, SDS–PAGE analyses to identify the elution peak *1 shown on the left. Protein constituents were determined by subsequent liquid chromatography-tandem mass spectrometry (LC-MS/MS) analysis.

