## [Peer Review File · The EMBO Journal]

Neurons dispose of hyperactive kinesin into glial cells for clearance

Chao Xie, Guanghan Chen, Ming Li, Peng Huang, Zhe Chen, Kexin Lei, Dong Li, Yuhe Wang, Augustine Cleetus, Mohamed Mohamed, Punam Sonar, Wei Feng, Zeynep Ökten, and Guangshuo Ou

Corresponding author: Guangshuo Ou (guangshuoou@mail.tsinghua.edu.cn)

Review Timeline:

Submission Date:	22nd Nov 23
Editorial Decision:	18th Dec 23
Revision Received:	2nd Feb 24
Editorial Decision:	23rd Feb 24
Revision Received:	6th Apr 24
Accepted:	25th Apr 24

Editor: Ieva Gailite

Transaction Report:

Dear Guangshuo,

Thank you for submitting your manuscript for consideration by the EMBO Journal. We have now received comments from three reviewers, which are included below for your information.

As you will see from the reports, all reviewers find the study of interest, while also pointing out several aspects that would need to be clarified in the final version before they can recommend acceptance of the manuscript. In particular, further characterisation of the exophers/extracellular vesicles involved in KIF5-CA removal would be needed, as indicated by reviewers #1 and #2. Based on the interest expressed in the reports, I would like to invite you to address the issues raised by the referees in a revised manuscript. I think it would be useful to discuss the revision in more detail via email or phone/videoconferencing - please let me know which option you prefer.

We generally allow three months as standard revision time. As a matter of policy, competing manuscripts published during this period will not negatively impact on our assessment of the conceptual advance presented by your study. However, please contact me as soon as possible upon publication of any related work to discuss the appropriate course of action. Should you foresee a problem in meeting this three-month deadline, please contact us to arrange an extension.

When preparing your letter of response to the referees' comments, please bear in mind that this will form part of the Review Process File and will therefore be available online to the community. For more details on our Transparent Editorial Process, please visit our website: <https://www.embopress.org/page/journal/14602075/authorguide#transparentprocess>. Please also see the attached instructions for further guidelines on preparation of the revised manuscript.

Please feel free to contact me if you have any further questions regarding the revision. Thank you for the opportunity to consider your work for publication. I look forward to your revision.

With best wishes,

Ieva

We realize that it is difficult to revise to a specific deadline. In the interest of protecting the conceptual advance provided by the work, we recommend a revision within 3 months (17th Mar 2024). Please discuss the revision progress ahead of this time with the editor if you require more time to complete the revisions.

Referee #1:

This is an ambitious paper regarding the extracellular release of kinesin motor proteins. The authors noticed that hyperactivated mutation of OSM-3 kinesin somehow resulted in hypomorphic mutation of this kinesin. Then they observed the dynamics using fluorescently labelled kinesin transgenic *C. elegans* sensory cilia, and found out that the extracellular release into "exophor"-like extracellular vesicles. They further performed FRET/FLIM analyses to confirm the conformational changes in vivo, suppressor screening to find Dyf-5 kinase as an intergenic suppressor of OSM3-CA, and finally showed that disease-causing KIF5-CA takes the similar fate, as some supporting information of the core finding.

Because the exophor release phenomenon of kinesin(s) itself is completely new, this paper holds a significant conceptual advance by itself. But the weak point is that authors stick to the *C. elegans* genetics and in vivo observation and that any cell biological supports for that those fluorescent dots really contain the fluorescently labeled mutant protein, because those extracellular structures can frequently hold nonspecific signals. Furthermore, characterization of the extracellular vesicles is very preliminary, and any double staining with known exophor markers and pharmacological intervention studies should be supplemented.

Accordingly, this reviewer will be glad to recommend its publication after improving the following points.

Major points:

- 1) The authors should carefully examine the specificity of the exophor fluorescent signals whether it really contains the mutant OSM3-CA protein, using immunofluorescence with appropriate negative controls and immunoblotting of extracellular vesicle fractions of the conditioned medium from transfected cells. Desirably, it is best to show if non-fluorescently-tagged OSM3-CA worm exophors do not give this much of fluorescence.
- 2) Also, please characterize those "exophor"-like structures by double staining with known markers and pharmacological studies against extracellular vesicle releasing pathways.
- 3) Does KIF5-CA overexpression result in similar exophor release from mammalian neurons? Is the released exophor labelled by immunoblotting? This is very important for the clinical interpretation of the authors' finding.

Minor points:

- 1) The authors should cite previous mammalian KO mouse studies of kinesins, for the introduction of the kinesin molecular motors.
- 2) It is very advisable if a careful time lapse study should be carried out to show how the peripheral kinesin accumulation is released as an exophor.

Referee #2:

The manuscript by Xie et al. describes fascinating phenomena related to the fate of the hyperactive kinesin protein, the ciliary kinesin OSM-3, which is disposed by axons from aberrant neurites. This is surprising, in that OSM-3 site of action in the cilium and OSM-3CA is located at the wrong end of the neuron. The authors demonstrated that the hyperactive OSM-3 disrupts cilia morphology, likely due to its absence in the cilia. Although OSM-3 forms a homodimer, the authors showed that the hyperactive

form of OSM-3CA does not exert a dominant negative effect on the normal protein; instead, it is recessive. This observation is consistent with the fact that the OSM-3CA form does not enter the cilia, but rather enters the axon and is shed from ectopic neurons. To confirm that the OSM-3CA form is extracellular, the authors used a myristoylated membrane marker, demonstrating that the puncta indeed co-localize with this marker. The authors then showed that the OSM-3CA form is taken up by glial cells and requires the PS receptor CED-1 for uptake. The end of the neurite co-localizes with actin and HSG-101, suggesting that the punctate shedding involves membrane scission. The authors subsequently discovered mechanisms for restoring cilia function in the OSM-3CA form through mutation in the MAPK DYF-5 and intramolecular mutation within OSM-3. They also demonstrated a conserved phenomenon of a hyperactive KIF5A being released extracellularly. These discoveries are fascinating, and the manuscript is generally well-written and will appeal to the broad readership of EMBO. However, there are three major concerns and some minor, all of which be addressed through revisions in writing and data analysis.

Major

1. Use of exopher vs extracellular vesicle vs axonal debris: Extracellular vesicles are remarkably diverse (refer to Figure 1 in this review by Buzas doi: 10.1038/s41577-022-00763-8) and cannot be categorized by size alone. Authors define extracellular OSM-3CA puncta as exophers based on a size larger than exosomes and microvesicles/ectosomes. Large exophers, identified by the Driscoll lab, are extruded from neuronal cell bodies, appear to function in "trash" elimination, and do not appear to share cytoskeletal features/biogenesis machinery (marked by TBB-2, actin, and ESCRT) with OSM-3CA extraneuronal puncta. Authors point out that axon debris is removed by ced-1 expressing engulfing cells, similar to OSM-3CA uptake by ced-1 expressing cephalic sheath cells. At this point, authors cannot conclude whether OSM-3CA is shed in EVs or axon debris. Given the lack of characterization of the contents and ultrastructural analysis of puncta associated with OSM-3CA, we strongly advise against use of the term "exopher" or "extracellular vesicle." Authors were judicious in their title and abstract wording, and should do so in the text.

2. OSM-3CA localizes to the axon, which, to our knowledge, is unipolar, microtubule plus end out (similar to cilia); dendrites are mixed polarity. In the introduction, authors write about ciliary proteins needing to move to the distal end of the dendrite, but do not consider how/why they are excluded from axons. That IFT reporters are still targeted to cilia in OSM-3CA mutants indicates that the sorting/trafficking/transport of ciliary proteins isn't disrupted. Why does the neuron dispose of OSM-3CA from only axons and not other parts of the cell?

3. The manuscript contains an enormous amount of data and represents a lot of work, on which authors are to be commended. However, the genetic suppressor section (p10-16) of the manuscript is dense, difficult to read, and disconnected from the abstract's main points. Streamlining will greatly improve the readability and increase the impact of the work. It would be great if authors could provide a model that integrates disposal/uptake, OSM-3CA intragenic suppressors, and OSM-3CA MAPK dyf-5 suppression.

Minor

1. The term "granule" is commonly associated with secretory vesicles. The authors' use of "granule" to describe OSM-3CA extracellular puncta should be revised for clarity and accuracy.
2. All statistical analyses conducted for multiple comparisons in the graphs utilize the Student's t-test, which is appropriate for comparing two datasets. For multiple comparisons involving one variant, a one-way ANOVA should be employed, or a two-way ANOVA for two variants.
3. In Figure 2F, the depiction of OSM-3CA aggregates in the axon being disposed of from neurites extending from the axon should be modified in the cartoon to more accurately convey this process.
4. On page 11, line 5, reconsider if "reinstated" accurately conveys the meaning of "reduced." Also, on page 11, line 23, the correct notation should be "H207QG444E."
5. That the R238W G/TGG allele was isolated 8 times suggests a "jackpot" mutation, same for dyf-5 L200H CT/AT (Supp. Figure 5). In the supplement, authors should provide more information on the screen - how many haploid genomes were screened?
6. Figure 1G: *osm-3(p802)* and *osm-3(sa125)* both display ectopic singlet microtubules, which is only mentioned in the figure legend but not the text. What is authors' interpretation?

Referee #3:

Xie et al., investigated the consequence of a mutation that renders the ciliary kinesin OSM-3 of *C. elegans* constitutively active. Specifically, they address why instead of behaving as a genetic gain-of-function, OSM-3CA behaves as a loss of function. They find that the hyperactive kinesin-1 is removed from the neuron in the form of "Exophers", which are taken up by the glial cells.

Uptake and elimination of OSM-3CA involves the engulfment receptor CED-1, which functions in the glia. The authors then provide a biophysical characterization of OSM-3CA and two intragenic mutations they identified. Through engineering of a rigor mutation (which suppresses the formation of the exopheres), they conclude that neurons can detect kinesin hyperactivation and target it to exophers. In addition, they identify that mutations in the ciliary kinase DYF-5 can suppress the ciliary phenotypes (but not exophers) of OSM-3CA, although it is not clear how this finding relates to the rest of the paper.

Overall, the observation made in the manuscript is interesting, and the experiments used to elucidate its underpinnings are largely done in a rigorous manner and with an impressive scope of approaches. If the comments below can be addressed, I would support publication.

Specific comments

- The authors' conclusion that the cell senses hyperactive kinesin is largely based on the use of a rigor mutation. However, since this mutation produces tight microtubule binding, it may be disrupting secretion by holding the kinesin firmly attached to the track. The authors should also test whether a mutation that disrupts microtubule binding suppresses exopher recruitment.
- At certain points in the discussion the authors acknowledge that OSM-3CA aggregates (which is consistent with its brightness), but this isn't clearly addressed. Can they do a FRAP on the aggregates in the neuronal cell body or axon, before the OSM-3CA is sequestered in the exopher?
- Figure 3E: the BafA and MG132 experiments are inconclusive. It is not clear what the effective concentration in the animal is, and it isn't clear if the effect is direct or indirect because both approaches give a subtle effect. Please either remove this panel (it has no real impact on the overall conclusion from the paper) or provide more conclusive experiments using mutations in proteasome/lysosomes components in glia.
- Please describe in the main text the experiments measuring ATPase activity. "Through biochemical investigations" is not informative.
- Figure 5 F,G: please add the positive control from the previous panels (OSM-3-GFP-mScarlet).
- Figure 6D: Please show the uncropped blot. Also, please provide the Coomassie of equivalent that shows the levels of dyf5 and osm3 constructs.
- Figure 6E: what is the klp-11 panel? Is it described in the main text?
- It was not clear to me what the contribution of Figure 6 is, since dyf5 mutants did not eliminate the exopheres. Can this be written in a clearer way?
- Figure 7: the data needs to be quantified, particularly since the representative images show a relatively mild effect. Also, please add zoomed-out images for panel A.
- Figure S7G: this is a quantification of OSM-3, so why is it in a figure about KIF5? It is also not described in the main text. Please remove from this figure and instead provide a quantification of the data in figure S7.

Referee #1:

This is an ambitious paper regarding the extracellular release of kinesin motor proteins. The authors noticed that hyperactivated mutation of OSM-3 kinesin somehow resulted in hypomorphic mutation of this kinesin. Then they observed the dynamics using fluorescently labelled kinesin transgenic *C. elegans* sensory cilia, and found out that the extracellular release into "exophor"-like extracellular vesicles. They further performed FRET/FLIM analyses to confirm the conformational changes in vivo, suppressor screening to find Dyf-5 kinase as an intergenic suppressor of OSM3-CA, and finally showed that disease-causing KIF5-CA takes the similar fate, as some supporting information of the core finding.

Because the exophor release phenomenon of kinesin(s) itself is completely new, this paper holds a significant conceptual advance by itself. But the weak point is that authors stick to the *C. elegans* genetics and in vivo observation and that any cell biological supports for that those fluorescent dots really contain the fluorescently labeled mutant protein, because those extracellular structures can frequently hold nonspecific signals. Furthermore, characterization of the extracellular vesicles is very preliminary, and any double staining with known exophor markers and pharmacological intervention studies should be supplemented.

We thank the reviewer for the positive assessment and constructive suggestion of our study.

Accordingly, this reviewer will be glad to recommend its publication after improving the following points.

Major points:

1) The authors should carefully examine the specificity of the exophor fluorescent signals whether it really contains the mutant OSM3-CA protein, using immunofluorescence with appropriate negative controls and immunoblotting of extracellular vesicle fractions of the conditioned medium from transfected cells. Desirably, it is best to show if non-fluorescently-tagged OSM3-CA worm exophors do not give this much of fluorescence.

We acknowledge the Referee's recommendation to examine the specificity of fluorescence attributed to OSM-3CA protein. However, performing immunofluorescence in *C. elegans* neurons poses significant challenges. In the 1990s, the Scholey lab developed antibodies against OSM-3 that were effective for Western blots and demonstrated ciliary localization in wild-type animals (Signor et al., *Mol Biol Cell*, 1999; PMID: 9950681). Yet, these antibodies predominantly accumulated in the ciliary middle segments during immunofluorescence, with minimal distal ciliary domain signals. This finding was perplexing, given OSM-3's role in distal segment construction and its movement along the entire length of cilia. Subsequently,

the Peterman lab's single-copy GFP insertion method for tracking OSM-3::GFP (Prevo et al., Nat Cell Biol, 2015; PMID: 26523365) revealed that OSM-3 predominantly occupies the distal segments, a result corroborated by our GFP knock-in animals (Xie et al., EMBO J, 2020). Despite extensive efforts, reliable immunofluorescence methodologies for adult *C. elegans* neurons, particularly for ciliary proteins, is currently unavailable.

The challenges extend to culturing *C. elegans* neurons. The protocol published two decades (Christensen et al., Neuron, 2002; PMID: 11856526) ago proved non-reproducible and inefficient, a sentiment echoed across various labs. Given these constraints, we propose deferring such experiments until a robust immunofluorescence method for *C. elegans* ciliated neurons becomes available.

On the other hand, our time-lapse imaging results of OSM-3CA::GFP puncta revealed consistent rotational motion within these structures (Movie S4). This observation, in conjunction with the fact that GFP alone does not demonstrate comparable rotational movement in puncta, strongly indicates that active motor proteins are present within OSM-3CA puncta. It also suggests that the green fluorescence observed in these dots is indeed emanating from the fluorescently tagged mutant protein, rather than being a result of GFP alone or nonspecific autofluorescence. We have included time-lapse recording frames and a movie to illustrate this phenomenon on the Figure 2G, EV Fig2C-D and Movie S1-3.

In response to the concern over not specific fluorescence (e.g., autofluorescence of the animal), we have now presented images from non-fluorescently-tagged OSM3-CA animals, in which no GFP fluorescence was detected, particularly in the nerve ring region where OSM-3CA::GFP puncta are localized. This should address the issue of potential non-specific fluorescence and further validate our findings. We have discussed this issue in the revised Discussion on page #19, line #13:

“...Future research will definitively ascertain whether GFP-tagged OSM-3CA puncta contain the hyperactive OSM-3 kinesin, pending the development of a reliable immunofluorescence protocol for *C. elegans* ciliated sensory neurons. Current time-lapse imaging of OSM-3CA::GFP puncta, as depicted in Fig. 2G, EV Fig2C-D and Movie S1-3, consistently exhibits rotational motion within these structures (Movie S4). This finding, combined with the lack of similar rotational movement in puncta with GFP alone, strongly suggests the presence of active motor proteins within these GFP-tagged OSM-3CA puncta. Moreover, imaging of non-fluorescently-tagged OSM3-CA animals revealed no detectable green fluorescence in the nerve ring region, where OSM-3CA::GFP puncta typically localize (EV Fig1C). These observations strongly indicate that the observed green fluorescence in the puncta originates from the fluorescently labeled mutant OSM-3CA protein, rather than GFP alone or nonspecific autofluorescence...”

2) Also, please characterize those "exophor"-like structures by double staining with known markers and pharmacological studies against extracellular vesicle releasing pathways.

After a thorough review of all publications concerning exophers since their initial discovery by Dr. Driscoll's lab in 2017, we have observed that exophers, as originally described, are large (approximately 4 μm) membrane-surrounded vesicles extruded by adult neurons in *Caenorhabditis elegans*. These exophers are known to contain protein aggregates and organelles, yet they have not been molecularly characterized, leaving their precise biomarkers unclear. Subsequent studies have not succeeded in isolating specific molecules unique to exophers. As a result, definitive markers for these vesicles remain unidentified. In addition, the process governing the release of exophers from the cell body remains an enigmatic aspect of cell biology, and to date, no pharmacological agent has been identified that can effectively disrupt or alter this exopher release mechanism. This represents a significant gap in our current understanding, and we plan to undertake the suggested experiments as soon as the necessary information becomes available.

Furthermore, we concur with Referee #2's comments regarding the inadvisability of using the terms "exophers" or "extracellular vesicles" without sufficient molecular characterization. We have accordingly revised our manuscript to reflect this understanding and have made the necessary corrections throughout the text.

We have now discussed this issue in the revised Discussion on page #8, line #7:

"...The measured diameter of the OSM-3CA::GFP puncta was approximately 1.4 μm , exceeding the size range of typical exosomes (30-100 nm in diameter) and other microvesicles (several hundred nanometers in diameter), but smaller than typical exophers, which have a diameter of 3.8 μm . (Melentijevic et al., 2017; Pegtel and Gould, 2019; van Niel et al., 2018). The particle size may appear to be a subtle distinction, and extracellular vesicles are remarkably diverse, regarding their different biogenesis and biological functions (Buzas, 2022). Large exophers, identified recently (Melentijevic et al., 2017), are extruded from neuronal cell bodies, appear to function in "trash" elimination, and do not appear to share cytoskeletal features or biogenesis machinery (marked by TBB-2, actin, and ESCRT) (EV Fig3) with OSM-3CA extraneuronal puncta and yet have not been molecularly characterized, leaving their precise biomarkers unclear. In addition, the process governing the release of exophers from the cell body remains an enigmatic aspect of cell biology, and to date, no pharmacological agent has been identified that can effectively disrupt or alter this exopher release mechanism. Given the lack of characterization of the contents and ultrastructural analysis of puncta associated with OSM-3CA, we henceforth refer to the ejected OSM-CA puncta as OSM-3CA ejecta..."

Exploring the use of pharmacological inhibitors to obstruct OSM-3CA release is indeed an important direction. As indicated in our manuscript, components of the ESCRT pathway are localized at the site of release, prompting us to search for potential ESCRT inhibitors. Unfortunately, our literature review and consultations with experts in the field of ESCRT and extracellular vesicles reveal a notable gap: to date, no specific inhibitors have been identified that can effectively target the ESCRT pathway in *C. elegans*, or even in mammalian systems. Thus, we may have to describe this lack of established inhibitor. The localization of ESCRT components in the release site suggests an involvement of the ESCRT pathway rather than an exopher-mediated process. We have discussed this issue in the revised text on page #8, line #3:

“...Furthermore, the future study will examine the contribution of the ESCRT pathway when an effective and specific pharmacological inhibitor targeting ESCRT is available...”

3) Does KIF5-CA overexpression result in similar exopher release from mammalian neurons? Is the released exopher labelled by immunoblotting? This is very important for the clinical interpretation of the authors' finding.

By examining a recent publication, we observed that KIF5-CA puncta are located externally to the neurons in which they are expressed, mirroring observations with hyperactive KIF17. These images were based upon GFP reporters. In discussions with Prof. Kristen Verhey, who conducted overexpression studies of hyperactive KIF17 in neurons (Hammond et al., *J Cell Biol.*, 2010; PMID: 20530208), it was confirmed that their cell culture conditions were meticulously managed, rendering the external KIF17 signal unlikely to be contaminants or derived from external sources. Prior to our study, the extracellular localization of these hyperactive kinesins was an unexplained phenomenon, not previously reported or discussed in the literature. We have included and cited the KIF5-CA and KIF17-CA images in the Appendix FigS2.

Our lab, focusing primarily on *C. elegans* research, plans to extend this inquiry through collaborations with experts in mammalian systems. We aim to investigate the release of hyperactive kinesin in cultured mammalian neurons and transgenic mice. This future research will be pivotal in translating our findings in *C. elegans* to clinical contexts, potentially offering new insights into the behavior and implications of hyperactive kinesins in neuronal systems. In the revised Discussion, we have included the Referee's suggestion to this direction on page #19 line #25:

“...While existing publications have provided evidence for the analogous release of hyperactive kinesins KIF5-CA and KIF17-CA in mammalian systems (Hammond et al., 2010; Pant et al., 2022), the precise documentation and reasons for their extracellular localization remain elusive. Moreover, these motor proteins have

predominantly been studied using GFP tags. Future investigations, therefore, should focus on examining endogenous hyperactive kinesins in mammalian neurons or patient samples, utilizing techniques like immunofluorescence or immunoblotting. Equally important is the development of live imaging system in mammalian neurons to reveal the mechanisms underlying the release of the pathogenic KIF5A-CA kinesin. Such studies are imperative for clinically interpreting our findings and understanding the broader implications of hyperactive kinesin behavior...”

Minor points:

1) The authors should cite previous mammalian KO mouse studies of kinesins, for the introduction of the kinesin molecular motors.

We have cited these references in our revised Introduction on page #2, line #21: “...regarding the essential roles of various kinesins in extensive biological processes illustrated in knock-out mice (Hirokawa et al., 2009; Midorikawa et al., 2006; Nonaka et al., 1998; Tanaka et al., 1998; Zhao et al., 2001) ...”

2) It is very advisable if a careful time lapse study should be carried out to show how the peripheral kinesin accumulation is released as an exophor.

We appreciate the advice. Our careful long-term time-lapse study has successfully demonstrated the process of OSM-3CA kinesin accumulation and its subsequent release in live *C. elegans*. These data have now presented in the Figure 2G, S2C-D and Movie S1-3. In the future, we will look for collaborators with expertise on mouse genetics. Developing an appropriate protocol and experimental system for these studies is essential for revealing the mechanisms underlying the release of peripheral pathogenic kinesin accumulations in mammalian systems. This future research will be instrumental in broadening our understanding of kinesin-related pathologies across different species. We have discussed this issue in the revised Discussion on page #19, line #31:

“...Equally important is the development of live imaging system in mammalian neurons to reveal the mechanisms underlying the release of the pathogenic KIF5A-CA kinesin...”

Referee #2:

The manuscript by Xie et al. describes fascinating phenomena related to the fate of the hyperactive kinesin protein, the ciliary kinesin OSM-3, which is disposed by axons from aberrant neurites. This is surprising, in that OSM-3 site of action in the cilium and OSM-3CA is located at the wrong end of the neuron. The authors demonstrated that the hyperactive OSM-3 disrupts cilia morphology, likely due to its

absence in the cilia. Although OSM-3 forms a homodimer, the authors showed that the hyperactive form of OSM-3CA does not exert a dominant negative effect on the normal protein; instead, it is recessive. This observation is consistent with the fact that the OSM-3CA form does not enter the cilia, but rather enters the axon and is shed from ectopic neurons. To confirm that the OSM-3CA form is extracellular, the authors used a myristoylated membrane marker, demonstrating that the puncta indeed co-localize with this marker. The authors then showed that the OSM-3CA form is taken up by glial cells and requires the PS receptor CED-1 for uptake. The end of the neurite co-localizes with actin and HSG-101, suggesting that the punctate shedding involves membrane scission. The authors subsequently discovered mechanisms for restoring cilia function in the OSM-3CA form through mutation in the MAPK DYF-5 and intramolecular mutation within OSM-3. They also demonstrated a conserved phenomenon of a hyperactive KIF5A being released extracellularly. These discoveries are fascinating, and the manuscript is generally well-written and will appeal to the broad readership of EMBO. However, there are three major concerns and some minor, all of which be addressed through revisions in writing and data analysis.

We appreciate that the Referee is also positive to our study.

Major

1. Use of exopher vs extracellular vesicle vs axonal debris: Extracellular vesicles are remarkably diverse (refer to Figure 1 in this review by Buzas doi: 10.1038/s41577-022-00763-8) and cannot be categorized by size alone. Authors define extracellular OSM-3CA puncta as exophers based on a size larger than exosomes and microvesicles/ectosomes. Large exophers, identified by the Driscoll lab, are extruded from neuronal cell bodies, appear to function in "trash" elimination, and do not appear to share cytoskeletal features/biogenesis machinery (marked by TBB-2, actin, and ESCRT) with OSM-3CA extraneuronal puncta. Authors point out that axon debris is removed by ced-1 expressing engulfing cells, similar to OSM-3CA uptake by ced-1 expressing cephalic sheath cells. At this point, authors cannot conclude whether OSM-3CA is shed in EVs or axon debris. Given the lack of characterization of the contents and ultrastructural analysis of puncta associated with OSM-3CA, we strongly advise against use of the term "exopher" or "extracellular vesicle." Authors were judicious in their title and abstract wording, and should do so in the text.

We concur with the Referee's suggestion on terminology. Accordingly, we have revised our manuscript to replace previous terms with "ejected OSM-CA puncta" or "OSM-CA ejecta." and discussed in the revised text on page #8, line #7:

"... The measured diameter of the OSM-3CA::GFP puncta was approximately 1.4 μm , exceeding the size range of typical exosomes (30-100 nm in diameter) and other microvesicles (several hundred nanometers in diameter), but smaller than typical

exophers, which have a diameter of 3.8 μm . (Melentijevic et al., 2017; Pegtel and Gould, 2019; van Niel et al., 2018). The particle size may appear to be a subtle distinction, and extracellular vesicles are remarkably diverse, regarding their different biogenesis and biological functions (Buzas, 2022). Large exophers, identified recently (Melentijevic et al., 2017), are extruded from neuronal cell bodies, appear to function in "trash" elimination, and do not appear to share cytoskeletal features or biogenesis machinery (marked by TBB-2, actin, and ESCRT) (EV Fig3) with OSM-3CA extraneuronal puncta and yet have not been molecularly characterized, leaving their precise biomarkers unclear. In addition, the process governing the release of exophers from the cell body remains an enigmatic aspect of cell biology, and to date, no pharmacological agent has been identified that can effectively disrupt or alter this exopher release mechanism. Given the lack of characterization of the contents and ultrastructural analysis of puncta associated with OSM-3CA, we henceforth refer to the ejected OSM-CA puncta as OSM-3CA ejecta..."

2. OSM-3CA locates to the axon, which, to our knowledge, is unipolar, microtubule plus end out (similar to cilia); dendrites are mixed polarity. In the introduction, authors write about ciliary proteins needing to move to the distal end of the dendrite, but do not consider how/why they are excluded from axons. That IFT reporters are still targeted to cilia in OSM-3CA mutants indicates that the sorting/trafficking/transport of ciliary proteins isn't disrupted. Why does the neuron dispose of OSM-3CA from only axons and not other parts of the cell?

We acknowledge the critical insights provided by previous studies, including our own research also published in EMBO Journal (Li et al. EMBO J, 2017; PMID: 28743734), which indicate that microtubule plus ends in the dendrites are oriented towards the cell body, facilitating transport by the minus-end-directed dynein-1 motor protein of ciliary components from the soma through dendrites to the ciliary base. Consequently, it is improbable that the plus-end-directed motor protein OSM-3CA kinesin would actively traverse along the dendrite. Given OSM-3CA's demonstrated capacity for microtubule-based movement in vitro, we hypothesize that OSM-3CA is more likely to travel into the axon, progressing toward the axon tip where microtubules predominantly exhibit plus-end orientation. Accumulation of excessive OSM-3CA at the axon tip might lead to its local release, potentially as a cellular response to the excess of this hyperactive motor protein.

We have included this in the revised Discussion on page #20, line #4:

"...Previous studies revealed that most of microtubules in the dendritic distal region are likely oriented towards the cell body, facilitating transport by the minus-end-directed dynein-1 motor protein of ciliary components from the soma through dendrites to the ciliary base (Li et al., 2017). Consequently, it is improbable that the plus-end-directed motor protein OSM-3CA kinesin would actively traverse

along the dendrite. Moreover, the recombinant OSM-3CA protein displayed no inclination towards in vitro condensation or aggregation, facilitating our analysis of its motility characteristics in vitro. We hypothesize that OSM-3CA is more likely to travel into the axon, progressing toward the axon tip where microtubules predominantly exhibit plus-end orientation (Li et al., 2017). Accumulation of excessive OSM-3CA at the axon tip might lead to its local release, potentially as a cellular response to the excess of this hyperactive motor protein...”

In response to Referee #2's second point, we have now incorporated a schematic model in our revised manuscript in Figure 7G. This model elucidates the transport dynamics of wild-type OSM-3 in its auto-inhibited state as it travels from the soma to the dendritic ending, serving as a cargo molecule. In contrast, we depict how the OSM-3CA variant might independently traverse along the axon.

3. The manuscript contains an enormous amount of data and represents a lot of work, on which authors are to be commended. However, the genetic suppressor section (p10-16) of the manuscript is dense, difficult to read, and disconnected from the abstract's main points. Streamlining will greatly improve the readability and increase the impact of the work. It would be great if authors could provide a model that integrates disposal/uptake, OSM-3CA intragenic suppressors, and OSM-3CA MAPK dyf-5 suppression.

We have moved the main figure of DYF-5 to supplements and provided an updated model figure (Fig. 7G) to summarize these events and regulations.

Minor

1. The term "granule" is commonly associated with secretory vesicles. The authors' use of "granule" to describe OSM-3CA extracellular puncta should be revised for clarity and accuracy.

We have made the correction by using “OSM-3CA ejecta” throughout the manuscript.

2. All statistical analyses conducted for multiple comparisons in the graphs utilize the Student's t-test, which is appropriate for comparing two datasets. For multiple comparisons involving one variant, a one-way ANOVA should be employed, or a two-way ANOVA for two variants.

We have made the corrections and provided the information for the type of statistic analyses in the corresponding figure legends.

3. In Figure 2F, the depiction of OSM-3CA aggregates in the axon being disposed of from neurites extending from the axon should be modified in the cartoon to more accurately convey this process.

We have provided a new cartoon to convey this process in the Figure 2F.

4. On page 11, line 5, reconsider if "reinstated" accurately conveys the meaning of "reduced." Also, on page 11, line 23, the correct notation should be "H207QG444E."

We have replaced "reinstated" with "reduced" and corrected the typo. We thank the Referee for pointing them out.

5. That the R238W G/TGG allele was isolated 8 times suggests a "jackpot" mutation, same for dyf-5 L200H CT/AT (Supp. Figure 5). In the supplement, authors should provide more information on the screen - how many haploid genomes were screened?

We have included the information in our revised manuscript on page #10, line #31:

"...Following an extensive screen of over 10^5 haploid genomes, we successfully isolated and cloned 22 suppressor mutants demonstrating DiI uptake comparable to that of wild-type organisms...."

6. Figure 1G: *osm-3(p802)* and *osm-3(sa125)* both display ectopic singlet microtubules, which is only mentioned in the figure legend but not the text. What is authors' interpretation?

We thank the Referee for their detailed examination of our data. Indeed, the figures demonstrate that the *osm-3(sa125)* strain, which carries the hyperactive OSM-3CA, exhibits a phenotype akin to that of the *osm-3(p802)* null allele. This finding, including the occurrence of ectopic singlet microtubules within the remaining middle segments, aligns with our previous findings published in Xie et al., EMBO J, 2020; PMID: 33433002. Our revised manuscript has now included a discussion of these aspects. In our early work, we discussed that the aberrant torque generation resulting from the loss of OSM-3 may disrupt the nine-fold symmetric arrangement of axonemal doublet microtubules with the insertion of ectopic singlets. (Xie et al., EMBO J, 2020; PMID: 33433002). We have incorporated relevant discussions in the revised manuscript on page #5, line #30:

"...Furthermore, we observed ectopic insertions of singlet microtubules interspersed among the doublet microtubules in the remaining middle segments of the *osm-3(sa125)* strain, which expresses hyperactive OSM-3CA (Fig. 1G). This phenotype closely resembles that of the *osm-3(p802)* null allele, as detailed in our earlier study (Xie et al., 2020). We propose that the aberrant torque generation resulting from the absence of OSM-3 may disrupt the axonemal structure's nine-fold

symmetric arrangement of doublet microtubules with the insertion of these ectopic singlets (Xie et al., 2020)”

Referee #3:

Xie et al., investigated the consequence of a mutation that renders the ciliary kinesin OSM-3 of *C. elegans* constitutively active. Specifically, they address why instead of behaving as a genetic gain-of-function, OSM-3CA behaves as a loss of function. They find that the hyperactive kinesin-1 is removed from the neuron in the form of "Exophers", which are taken up by the glial cells. Uptake and elimination of OSM-3CA involves the engulfment receptor CED-1, which functions in the glia. The authors then provide a biophysical characterization of OSM-3CA and two intragenic mutations they identified. Through engineering of a rigor mutation (which suppresses the formation of the exophers), they conclude that neurons can detect kinesin hyperactivation and target it to exophers. In addition, they identify that mutations in the ciliary kinase DYF-5 can suppress the ciliary phenotypes (but not exophers) of OSM-3CA, although it is not clear how this finding relates to the rest of the paper. Overall, the observation made in the manuscript is interesting, and the experiments used to elucidate its underpinnings are largely done in a rigorous manner and with an impressive scope of approaches. If the comments below can be addressed, I would support publication.

We are glad that the Referee appreciated our efforts and is also constructive to our study.

Specific comments

- The authors' conclusion that the cell senses hyperactive kinesin is largely based on the use of a rigor mutation. However, since this mutation produces tight microtubule binding, it may be disrupting secretion by holding the kinesin firmly attached to the track. The authors should also test whether a mutation that disrupts microtubule binding suppresses exopher recruitment.

We recently identified a mutation, H207Q, which impairs microtubule binding as evidenced by our latest microtubule gliding assay. Intriguingly, the H207Q mutation reduces the ejection of OSM-3CA and successfully restores ciliary structure in the OSM-3CA mutant background. This novel finding is included in our revised manuscript on page #16, line #23:

“...The rigor mutation G235A potentially shields OSM-3CA from its predisposition to aggregation through robust microtubule binding. Conversely, our findings show that the H207Q mutation in OSM-3 fails to bind microtubules, as evidenced by the absence of microtubule binding in the microtubule gliding assay (Fig. 5G) and minimal microtubule-stimulated ATPase activity (Fig. 5F). Notably, worms with

either the *osm-3(H207Q)* mutation or the *osm-3(H207Q;G444E)* combination, derived from genetic screening or knock-in methods (Fig. 5C, EV Fig5D, EV Fig5F), exhibited no punctate signals. Instead, their distribution was dispersed in the cilia and soma of ciliated neurons (Fig. 5C, EV Fig5D, EV Fig5F). Given that the H207Q mutation results in a loss of ATPase activity without the heightened microtubule-binding capability observed in the G235A mutant, it implies that the ejection of OSM-3CA puncta is primarily driven by hyperactivity...”

- At certain points in the discussion the authors acknowledge that OSM-3CA aggregates (which is consistent with its brightness), but this isn't clearly addressed. Can they do a FRAP on the aggregates in the neuronal cell body or axon, before the OSM-3CA is sequestered in the exopher?

We performed the suggested FRAP experiments and added a new figure and section in our manuscript. We believe that these new results further improve our study and appreciate the advice from the Reviewer.

Dynamics of OSM-3CA within ejecta and neurites

Our fluorescence recovery after photobleaching (FRAP) experiments shed light on the dynamic nature of OSM-3CA within ejecta. When we photobleached a segment of the OSM-3CA ejecta, characterized by rotational movement, we observed a remarkably rapid recovery half-time of just 0.26 seconds, with the recovery level reaching 47.3% (Fig. 3A-B). This finding emphasizes that OSM-3CA ejecta are not static protein aggregates but rather dynamic structures. This dynamic recovery is likely facilitated by the movement of OSM-3CA along microtubules or its diffusion from unbleached regions, as suggested by the filamentous TBB-2 signals within the OSM-3CA ejecta and its filament-like movements (Movie S4, EV Fig3H).

To delve deeper into the process of OSM-3CA incorporation into ejecta from neurites, we conducted full-area bleaching on OSM-3CA ejecta connected to such neurites. Tracking fluorescence changes in both the ejecta and neurites revealed a significantly lengthened recovery half-time of 5.0 seconds in the ejecta, in stark contrast to the 0.26 seconds observed in partial bleach experiments (Fig. 3C-E). This extended recovery period mirrors the fluorescence decay in the neurites, indicating a continuous and rapid accumulation of hyperactive OSM-3CA in the ejecta.

However, when we subjected the entire OSM-3CA ejecta detached from the neurites to bleaching, we observed a minimal fluorescence recovery of only 3.9% (Fig. 3F-G). This limited recovery is consistent with the expectation that no new OSM-3CA protein influx would occur after the ejecta's detachment from the neuron, where OSM-3CA is originally produced.

- Figure 3E: the BafA and MG132 experiments are inconclusive. It is not clear what the effective concentration in the animal is, and it isn't clear if the effect is direct or

indirect because both approaches give a subtle effect. Please either remove this panel (it has no real impact on the overall conclusion from the paper) or provide more conclusive experiments using mutations in proteasome/lysosomes components in glia.

In accordance with the published protocol, we utilized both drugs in our study. However, we agree with the Reviewer's suggestion that their inclusion had little real impact on the overall conclusion from the paper. Therefore, we followed the Reviewer's advice and omit these results from our manuscript.

- Please describe in the main text the experiments measuring ATPase activity. "Through biochemical investigations" is not informative.

We have described in the main text:

... Through microtubule-stimulated ATPase activity assays, we established that H207Q and R238W markedly attenuated ATPase activity to 18% and 39% comparing to the wild-type OSM-3, respectively (Fig. 5F). Furthermore, H207Q; G444E and R238W; G444E double mutants reduced the hyperactivity exhibited by OSM-3G444E which is over three times higher than that of wild type OSM-3, to a level even lower than the wild type but higher than that conferred by each individual mutation (Fig. 5F).

- Figure 5 F,G: please add the positive control from the previous panels (OSM-3-GFP-mScarlet).

We have added the controls in the new Figure 5 F and G.

- Figure 6D: Please show the uncropped blot. Also, please provide the Coomassie of equivalent that shows the levels of dyf5 and osm3 constructs.

In the revised Appendix FigS1D, we have now included the Coomassie-stained image of DYF-5 and OSM-3 tail protein preparations. These samples were processed in 2018 by Zeynep Okten's laboratory at the Technische Universität München. While the Okten lab did not have an SDS-PAGE for the radioactive gel due to equipment limitations, the lanes visible in the radiographs were obtained from the purification process shown in the cropped lane, which was subsequently divided into three equal volumes and phosphorylated with DYF-5, as also depicted in the cropped gel.

Furthermore, the Ou lab conducted mass spectrometry analysis of the in vitro phosphorylation reaction described above at Tsinghua University. This analysis successfully identified S658, S662, and T666 on the OSM-3 tail as the phosphorylation sites. We are currently conducting additional experiments to gain a deeper understanding of the functional significance associated with the phosphorylation of the OSM-3 tail by the DYF-5 kinase.

- Figure 6E: what is the klp-11 panel? Is it described in the main text?

We have described and discussed this in the revised main text on page #16, line #8:

“...To investigate the role of heterotrimeric kinesin-II in the ciliary localization of GFP-tagged OSM-3G235A, particularly in the remaining middle segment, we introduced a klp-11 null allele, which impairs kinesin-II function, into the osm-3(G235A) strain. Our observations revealed that OSM-3G235A was absent from the cilia but accumulated at the ciliary base (Fig. 7A). This finding suggests that the observed presence of OSM-3G235A in the middle segment is not due to the inherent movement of OSM-3G235A itself but rather results from the import by kinesin-II....”

- It was not clear to me what the contribution of Figure 6 is, since dyf5 mutants did not eliminate the exopheres. Can this be written in a clearer way?

We have clarified this issue in the revised text on page #15, line #16:

“...Although the osm-3CA intragenic suppressors completely eliminated OSM-3CA::GFP ejecta and restored the ciliary localization of OSM-3CA::GFP (Fig. 5C, S5F), the reduction of OSM-3CA::GFP ejecta was not evident in osm-3CA; dyf-5 double mutants. However, a small proportion of OSM-3CA::GFP was restored within the cilia, despite its immobility (Appendix Fig. S1B, S1F). Our in vitro protein kinase assays revealed that DYF-5 directly phosphorylates the C-terminal fragment of OSM-3 (444-699) (Appendix Fig. S1D-E), suggesting that DYF-5 kinase may act directly on OSM-3CA. Additionally, DYF-5's ability to phosphorylate other IFT proteins (Jiang et al., 2022) indicates potential alternative mechanisms. It is conceivable that the absence of DYF-5-mediated phosphorylation could alter OSM-3 CA's conformation and activity, either directly or indirectly, thereby partially recovering its localization in the cilia...”

We have also moved the Figure 6 to Appendix FigS1.

- Figure 7: the data needs to be quantified, particularly since the representative images show a relatively mild effect. Also, please add zoomed-out images for panel A.

We have provided quantifications as Figure 7F and zoomed-out images for panel A.

- Figure S7G: this is a quantification of OSM-3, so why is it in a figure about KIF5? It is also not described in the main text. Please remove from this figure and instead provide a quantification of the data in figure S7.

We have provided quantification for this figure in Appendix FigS2 and removed the former Fig. S7G.

Dear Guangshuo,

Thank you for submitting a revised version of your manuscript. Your study has now been seen by all original referees. Based on the positive assessment of reviewers #2 and #3 and their further input on the limited availability of tools for identification of specific extracellular compartments in *C. elegans*, I will accept the manuscript after the editorial points listed below have been addressed.

1. Please submit up to five keywords.
2. Email for Punam Sonar (punam.sonar@tum.de) did not reach the addressee, please check and correct.
3. Please check that the funding information is correct, complete and identical both in the manuscript and our online system.
4. Please submit a complete author checklist, which you can download from our author guidelines (<https://www.embopress.org/pb-assets/embo-site/EMBO%20Press%20Author%20Checklist-1642513524327.xlsx>). Please insert information in the checklist that is also reflected in the manuscript. The completed author checklist will also be part of the Review Process File.
5. Please make sure that the order of the sections in the manuscript is as follows: abstract, introduction, results, discussion, materials & methods, data availability section, acknowledgments, disclosure statement and competing interests, references, main figure legends, tables, expanded figure legends.
6. CRedit has replaced the traditional author contributions section because it offers a systematic, machine-readable author contributions format that allows for more effective research assessment. Please remove the Authors Contributions from the manuscript and use the free text boxes beneath each contributing author's name in our online submission system to add specific details on the author's contribution. More information is available in our guide to authors.
7. Please rename "Conflict of interest" section into "Disclosure and competing interests statement" (further info: <https://www.embopress.org/page/journal/14602075/authorguide#conflictsofinterest>).
8. Please move the Data Availability section to the end of Materials and Methods. As far as I can see, no data deposition in external databases is needed for this paper. If I am correct, then please state in this section: This study includes no data deposited in external repositories. Further information can be found at <https://www.embopress.org/page/journal/14602075/authorguide#dataavailability>
9. Please rename the movies into Movie EV1-EV8 and update the callouts accordingly. The legends should be removed from the Appendix file and zipped with each movie file. Further information is available here: <https://www.embopress.org/page/journal/14602075/authorguide#expandedview>
10. Please move Appendix methods to the main manuscript text file.
11. Please update the Appendix table callouts in the methods section - the nomenclature should be Appendix Table S1-S3.
12. In our standard image check, we noticed image reuse between the panels indicated below. Please mention the reuse in the corresponding figure legends:
 - Image re-use between Figure 1F and Appendix Figure S1B.
 - Image re-use between Figure 2C and Figure 7C.
 - Image re-use between Figure 5C and Figure 7A.
 - Image re-use between Figure 6B and Figure 6D.
 - Image re-use between Figure 7A and Figure EV Fig 5C.
13. Source data for Fig 1C are missing, please submit the corresponding file.
14. Our data editors have flagged the following issues in figure legends that need correcting:
 - The legends for figures EV 4b-e are not provided in the sequential manner (legend for figures EV 4c, e is provided before legend of figures EV 4b, d, respectively). This needs to be rectified.
 - The legends for figures EV1f-h is incorrectly labelled as EV1e-g. This needs to be rectified.
 - Please note that in figure EV 4b there is a mismatch between the annotated p values in the figure legend and the annotated p values in the figure file that should be corrected.
 - The box plot needs to be defined in terms of minima, maxima, centre, bounds of box and whiskers, and percentile in the legend of figure 7f.
 - The information related to n is missing in the legends of figure 7f; EV 4b, d, f.
 - Although 'n' is provided, please describe the nature of entity for 'n' in the legends of figures 1i; 5d; EV 3d, g.
 - The error bars are not defined in the legends of figures EV 4b, d, f.
 - The scale bar is missing for figures EV 1d-e; EV 2a.
 - The scale bar needs to be defined for figures EV 1c, f; EV 3b, e.
 - The scale bar and its definition are missing for figures EV 3c, f.
 - The yellow arrow is not defined in the legend of figures EV 3c, f.
15. Papers published in The EMBO Journal are accompanied online by a 'Synopsis' to enhance discoverability of the manuscript. It consists of A) a short (1-2 sentences) summary of the findings and their significance, B) 3-4 bullet points highlighting key results and C) a synopsis image that is 550x300-600 pixels large (width x height, jpeg or png format). You can either show a model or key data in the synopsis image. Please note that the image size is rather small and that text needs to be readable at the final size. Please send us this information together with the revised manuscript.

Please let me know if you have any questions regarding any of these points. You can use the link below to upload the revised

files.

With best wishes,

Ieva

We realize that it is difficult to revise to a specific deadline. In the interest of protecting the conceptual advance provided by the work, we recommend a revision within 3 months (23rd May 2024). Please discuss the revision progress ahead of this time with the editor if you require more time to complete the revisions.

Referee #1:

In this revision the authors have supplemented the negative controls according to this and other reviewers' suggestions, which considerably improved the quality of data presentation and specificity. However, as this reviewer has also pointed out in the previous review, the precise cell biological (=biochemical and morphological) identification of the "exophor" should be accomplished for its publication on this class of journals. Also, the neuronal degeneration argument should be based on multidisciplinary studies involving other experimental systems including mammalian neurons, which this reviewer had strongly recommended. Accordingly, because of the lack of deep materialistic identification of the proposed structure, it would be suitable for more specific journals only with the current dataset.

Referee #2:

Authors did an amazingly thorough and convincing job of addressing all of my previous concerns and those of other reviewers. This included new data, revisions to text, and addition of cartoons/models to increase impact of the manuscript. Thank you!

Referee #3:

The authors have done a good job addressing my comments and I support publication. The paper is really nice - congrats!

The authors addressed the minor editorial issues.

Dear Guangshuo,

Thank you for addressing the final points. I sincerely apologise for the delay in communicating the decision due to the high number of submissions we receive and the busy conference schedule at the moment. I am now pleased to inform you that your manuscript has been accepted for publication. Congratulations on a nice study!

Before we forward your manuscript to our publishers, there are a couple of points that need to be addressed:

- 1) There is a typo in the provided synopsis image - should be "glia" instead of "gilia" - please update the file.
- 2) The source data for figure panel 1C consist of the figure panel. We would need the numerical data used to create the graph. You can send me the updated source data via email.
- 3) I would like to propose some minor edits in the manuscript abstract and synopsis (please also see the attached file). I have also written a short blurb that will accompany the title of your manuscript in our online system. Please let me know if any corrections are needed:

Blurb:

Phagocytic receptor CED-1 enables glial uptake and subsequent degradation of hyperactive kinesins in *C. elegans*.

Synopsis:

Hyperactivation of kinesins has been linked to neurodegenerative conditions. This study shows that a hyperactive version of OSM-3, a member of kinesin-2 superfamily that drives anterograde intraflagellar transport in *C. elegans*, is expelled from neurons for degradation in glial cells.

- Hyperactive OSM-3CA kinesin is absent from cilia and forms dynamic puncta within the neurites of sensory neurons.
- OSM-3CA puncta are ejected from sensory neurons and engulfed by neighboring cephalic sheath glia cells in a manner dependent of the phagocytic receptor CED-1.
- Mutations that reduce OSM-3CA activity or increase its microtubule binding protect OSM-3CA and restore normal cilia.
- ALS-linked hyperactive kinesin-1 is also expelled from neurons when ectopically expressed in *C. elegans*.

If you have any questions, please do not hesitate to contact the Editorial Office. Thank you for this contribution to The EMBO Journal and congratulations on a successful publication!

With best wishes,

leva

leva Gailite, PhD
Senior Scientific Editor
The EMBO Journal
Meyerohofstrasse 1
D-69117 Heidelberg
Tel: +4962218891309
i.gailite@embojournal.org

>>> Please note that it is The EMBO Journal policy for the transcript of the editorial process (containing referee reports and your

response letter) to be published as an online supplement to each paper. If you do NOT want this, you will need to inform the Editorial Office via email immediately. More information is available here: https://www.embopress.org/transparent-process#Review_Process
